# The impact of hyperlinks, skim reading and perceived importance when reading on the Web

**Lewis T. Jayes**[1]☯, **Gemma Fitzsimmons**[2]☯*, **Mark J. Weal**[3]☯, **Johanna K. Kaakinen**[4,5]☯, **Denis Drieghe**[2]☯

**1** School of Psychology, University of Surrey, Guildford, United Kingdom, **2** School of Psychology, University of Southampton, Southampton, United Kingdom, **3** School of Electronics and Computer Science, University of Southampton, Southampton, United Kingdom, **4** Department of Psychology, Finland INVEST Research, University of Turku, Turku, Finland, **5** Flagship, University of Turku, Turku, Finland

☯ These authors contributed equally to this work.
* gemma.fitzsimmons@soton.ac.uk

**Data Availability Statement:** The data underlying the results presented in the experiments in this manuscript are available from the UK Data Service.

## Abstract

It has previously been shown that readers spend a great deal of time skim reading on the Web and that this type of reading can affect comprehension of text. Across two experiments, we examine how hyperlinks influence perceived importance of sentences and how perceived importance in turn affects reading behaviour. In Experiment 1, participants rated the importance of sentences across passages of Wikipedia text. In Experiment 2, a different set of participants read these passages while their eye movements were tracked, with the task being either reading for comprehension or skim reading. Reading times of sentences were analysed in relation to the type of task and the importance ratings from Experiment 1. Results from Experiment 1 show readers rated sentences without hyperlinks as being of less importance than sentences that did feature hyperlinks, and this effect is larger when sentences are lower on the page. It was also found that short sentences with more links were rated as more important, but only when they were presented at the top of the page. Long sentences with more links were rated as more important regardless of their position on the page. In Experiment 2, higher importance scores resulted in longer sentence reading times, measured as fixation durations. When skim reading, however, importance ratings had a lesser impact on online reading behaviour than when reading for comprehension. We suggest readers are less able to establish the importance of a sentence when skim reading, even though importance could have been assessed by information that would be fairly easy to extract (i.e. presence of hyperlinks, length of sentences, and position on the screen).

## Introduction

Current research has consistently shown that reading on the Web differs from reading in other contexts. One specific difference is the presence of hyperlinks, words that enable users to navigate to a different webpage when clicked. Hyperlinks are visually salient and have been

The DOI is: https://dx.doi.org/10.5255/UKDA-SN-855044.

**Funding:** GF was funded by an EPSRC grant for the Doctoral Training Centre in Web Science: EP/G036926/1. This work formed a part of a PhD completed in the Web Science DTC. The funders had no role in study design, data collection and analysis, decision to publish, or preparation of the manuscript.

**Competing interests:** The authors have declared that no competing interests exist.

shown to anchor attention during reading on the Web [1]. Another difference is the fact that there is so much information on the Web, that it is often not considered viable to read all available information for comprehension. As a result of this, readers will often not consider all text to be of equal importance, depending on their task. Skim reading is one common adaptive reading behaviour that is adopted in order to make reading on the Web more manageable for the reader. Indeed, it has previously been noted that screen-based reading behaviour is more likely to be characterised by 'more time spent browsing and scanning, keyword spotting . . . non-linear reading, and reading more selectively' [2]. Further study has found online reading is conducted more quickly, with a cost to comprehension [3,4]. Additionally, education research suggests students need to be taught to not over rely on scanning behaviour, due to the cost to processing incurred [5]. The increase of skimming in online reading behaviour therefore invites the question of how readers assess the importance of text, when engaged in a reading strategy that is so rapid and selective ([2,6,7]). As such, this research aims to explore how aspects of reading on the Web, specifically skim reading, the presence of hyperlinks and the composition of a webpage affect a readers' perceived importance of the text and in turn their reading behaviour.

## Reading on the web and text importance

Eye tracking is a valuable method for investigating digital reading as it provides a moment-to-moment record of how readers allocate attention when viewing webpages [8,9]. Primarily, eye movements measures reveal where readers are allocating attention during reading and how long is spent doing this, via the length of fixations on words. As such, eye tracking measures reflect the time course of how readers move word-by-word through webpages [1,10]. Eye tracking measures typically consider the initial duration of fixation on words, how often regressive eye movements occur (movements towards previous text) and the amount of re-reading of words (see [11] for review). Each are taken as an index of the degree of online lexical processing required for readers to comprehend the text [12,13]. As such, a number of studies have investigated key differences between eye movement reading patterns and reading on the Web. Typically, these explore the processing costs of hyperlinks and how they affect attention allocation when reading in a Web environment.

Using eye tracking, Fitzsimmons, Weal and Drieghe [14] demonstrated that the presence of hyperlinks themselves (i.e. blue words) does not have a negative impact on reading, in a static environment (i.e. the hyperlinks could not be clicked to open new webpages) resembling a Wikipedia page, when reading for comprehension. Early fixation measures and skipping probability were not affected by whether the word was a hyperlink or not. The only difference observed was that linked words were more likely to induce re-reading, but only if the hyperlinked word was also low frequency. Converging evidence from Gagl [15] also found no perceptual disadvantage from the use of hyperlinks, in terms of fixation durations and skipping rates, despite a lack of blue light sensitive retinal cells in the centre of the fovea. In addition to no disadvantages to foveal and parafoveal perceptibility, it was also found that blue or underlined words (which were on average rather low-frequency) were more likely to be re-read, suggesting increased attraction of attentional resources during re-reading of hypertext. It could be argued this is due to their perceived importance when reviewing the content of a sentence.

Fitzsimmons, Jayes, Weal and Drieghe [1] extended this investigation by introducing the task of having participants either read the Wikipedia page for comprehension or skim read the text. When the text was presented as a static webpage, readers were shown to fully lexically process all words when reading for comprehension. When skim reading, however, only linked words were shown to be fully processed. The fact that unlinked words were not fully processed

was evidenced by a lack of frequency effect, whereby low frequency words are typically fixated for longer than high frequency words, an effect which was present for the linked words [16–18]. These findings suggested readers were prioritising the processing of visually salient words while skim reading webpages. When the task of navigation was introduced (i.e. the ability to click on a link to open a new webpage), once again only linked words were fully lexically processed. However, this was the case regardless of whether readers were asked to skim read or read for comprehension. Both these studies support the notion that the Web environment, and its additional physical and task differences, affect reading behaviour. Specifically, when skim reading or navigating, readers use links to 'anchor' their attention, and these links are used to guide the reader through the text in as efficient a manner as possible. The nature of the Web and the composition of Webpages encourages a reading strategy whereby readers prioritise visually salient information, in order to read through Webpages quickly.

Like hyperlinks, other typographical cues, such as boldface or underline, have also been shown to serve to highlight a word or small section of text. Research has shown that making a keyword or phrase distinct in the text results in the reader paying more attention to the emphasised content when reading [19] and often results in better memory for those emphasised pieces of text [19–22], as well as accelerated lexical processing [23]. Similarly, hyperlinks are words or phrases in the text that readers treat differently to the other words in the text, partly due to their increased visual salience.

In addition to the topographical differences reading on the Web carries, the increased likelihood of skim reading on the Web [2] is another key difference compared to other reading activities. Furthermore, there is a great deal of evidence suggesting that during skim reading, some comprehension may be lost compared to more in-depth reading [24–27]. This loss in comprehension is not, however, consistent across all the text being read. There appears to be a difference between information regarded as important versus unimportant. Previous research has specifically shown that information viewed as important (as rated by independent participants) does not appear to receive the same loss of comprehension observed for unimportant information [7,27,28]. To explain these findings, it has been suggested that readers engage in an adaptive strategy in order to gain as much information from the text as possible, in a reduced time. But in order to do this the reader must judge which pieces of information are important and which pieces are not. When reading text, it may be difficult to judge how important something is until after you have read and understood it. However, there might be specific cues that the reader can use to predict importance, especially when reading on the Web in a hypertext environment, such as hyperlinks. This is supported by evidence showing that judgements of text importance are often affected by typographical signals, rather than by semantic content. Lorch, Lemarié and Grant [29] found the use of asterisks to demarcate information could cause text of less semantic importance to be perceived being more important. This suggests that signals can override semantic cues from the text. This means participants are willing to use signalling rather than make their own judgements based on the semantic content of the text. It is therefore possible that the presence of hyperlinks within text may be used as a signal of importance due to their visual salience.

As such, this study focuses on how the reader might use cues in the text, such as hyperlinks, to suggest where important information might lie in the text. In Experiment 1, we asked readers to judge the importance of sentences within passages of text (in this case, edited Wikipedia articles). It would be reasonable to assume that hyperlinks could be viewed in a similar way to other signals and thus render the hyperlinked text more influential to participants' perception of importance independent of the semantic content of the text [29]. In Experiment 2, we additionally explored how the task of skimming versus reading for comprehension affected the relationship between reading and importance of text. It has previously been shown that

reading times are increased for important text [19]. It is possible, however, given the effect skimming has on reducing lexical processing of unlinked hypertext [1], that skim reading could reduce the influence of importance as readers are less able to establish this aspect of the reading process. As such, the current research also aimed to provide suggestions for the optimal presentation of information to readers, given the prevalence of hyperlinks and increased proclivity for skim reading on the Web.

## Experiment 1

In Experiment 1, we investigated the impact of hyperlinks and composition of webpages on importance judgements of text. In order to measure this, we conducted Experiment 1 with two groups. The first group viewed passages of text in a Wikipedia environment. Links were present in the text and the participants had to rate each sentence for its overall importance in that passage. In the second group, participants saw the same passages of text in a reduced Wikipedia environment, with all embedded links throughout all the passages removed. In both groups, the text they read was presented on the screen which also featured the Wikipedia logo, search bar and the sidebar articles (example stimuli available: https://goo.gl/JLvvMD). By doing this, we were able to separate the impact of hyperlinks from the importance of other textual and content factors. We predicted that, in the absence of hyperlinks to signal importance, the semantic content will be primarily used to judge importance, as suggested by Lorch, Lemarié and Grant [29]. In the case of hyperlinked text, signalling research suggests that if a typographical signal is present, they result in readers paying more attention to the emphasised content. As such, we predicted sentences featuring hyperlinks to be rated as more important than sentences without hyperlinks.

Two additional factors investigated in Experiment 1 were the length of the sentence and the sentence position on the page. Firstly, we predicted that longer sentences would be rated as more important than shorter sentences due to the so-called information bias. Information bias is the belief that the more information that can be acquired to make a decision, the better, even if that extra information is irrelevant for the decision (e.g. [30]). So regardless of the semantic content of a sentence, if the sentence is longer the participant will be biased to think it must be important because there is more information. Secondly, we predicted sentences that appear higher up the webpage would be rated as more important than sentences lower down the page. A common mass media writing style is to write articles with the most important information at the top and the least important at the bottom, known as the "inverted pyramid" structure [31]. Essential information is included in the lead paragraph. Additional details, background, or other information are typically added to the article in order of importance, such that the least important items are at the bottom. The inverted pyramid originates from old media technology such as the telegraph, whereby the most important information was always transmitted first [31]. It remains a very common media writing style and it is reasonable to assume that it builds a prior set of expectations to the readers when they are reading articles. Furthermore, studies analysing gaze patterns have shown the eye movement behaviour tends to follow this pattern [32].

### Method

**Participants.**  Fifty native English speakers (Linked Experiment: 3 male, 29 females, with an average age of 20.22 years; Unlinked Experiment: 2 male, 16 females, with an average age of 20.33 years) participated in exchange for course credits or payment (£6) and were members of the University of Southampton community, predominately Psychology undergraduate students. All had no known reading difficulties. None of the participants took part in Experiment

2. The imbalance in participant numbers between the linked and unlinked experiments was due to the respective analyses carried out on these datasets. The unlinked group was used as a point of comparison with the linked group, while the linked group was further analysed with additional manipulations to test the effect of number of links, position on page and sentence length. As such, more participants were required for the analysis of the linked dataset. All sample sizes reported in this manuscript were chosen to be comparable to sample sizes used in previous research exploring the effects of the Web on reading [1,10,33,34] and are typical of eye movement and reading studies. In addition, post hoc calculations of power were conducted given the current sample size using the simr package in R [35] and consistently returned an estimated power above 80% with the significance level of $\alpha$ = .05 (as suggested by Cohen [36]). Across the analyses for Experiment 1, power values ranged from 80–100% for main effects (effect sizes ranged between .01-.28).

**Apparatus.** Participants were seated in a cubicle in front of a desktop computer monitor and a laptop computer. The desktop computer was used to present each edited Wikipedia article in its entirety. Meanwhile, the laptop displayed each sentence from the edited Wikipedia article individually. The sentences appeared one at a time on the laptop screen in the same order as the edited Wikipedia article shown on the desktop computer monitor. This setup allowed participants to rate the importance of individual sentences while still seeing how it fits with the rest of the passage. Sentences were presented in 14pt mono-spaced Courier font.

**Stimuli and design.** The stimuli in Experiment 1 consisted of forty edited Wikipedia articles (example stimuli available: https://goo.gl/JLvvMD) taken from Experiment Three of Fitzsimmons et al. [10]. The Wikipedia articles were ten to twelve lines in length (between 118–162 words in length). The text within the Wikipedia articles was identical to the source material, at the time it was sampled, with the exception of four sentences per stimulus. These sentences were embedded into the original Wikipedia articles, amounting to four per article. In Fitzsimmons et al., [1] one hundred and sixty words were embedded in these experimental sentences (one word per sentence) and these experimental sentences were designed to be semantically consistent with the text already present, so as not to stand out from the existing text. This decision was made so that the articles were as close to a natural Web environment as possible, while featuring the additional experimental sentences. All original links were also retained in the text (between 1–3 words).

The three continuous independent variables considered in Experiment 1 were number of links in a sentence (all links were a single word), position on the screen of sentence (line number) and length of sentence (in number of words). In addition, we used the categorical, between-subjects independent variable of Passage Type (Passages with Links vs. Passages without Links) to compare ratings between the Linked group and Unlinked group. This was employed to investigate the effect of hyperlinks themselves on how readers interpret the importance of text. This variable was implemented between subjects, as we did not want the ratings given to the unlinked sentences to be influenced by the presence of hyperlinks in other sentences during the same experiment.

In the original Fitzsimmons et al. [10] study, four versions of each (un)linked word were produced based on the variables of Word Frequency (high vs. low) and Word Type (Linked vs. Unlinked) which were 4–7 characters in length (average length = 5.24 characters). The high frequency words had an average log transformed HAL frequency of 9.91 (range 8.13–12.66). and the low frequency words has an average log transformed HAL frequency of 5.75 (range 2.77–9.35). This was a within-subjects design, with the 4 different inserted word types rotated according to a Latin Square. Word Frequency and Word Type were not the subject of interest in this article (see [1] for the analysis of these variables), but we did present the stimuli according to the same Latin square, meaning every participant saw only one version of each edited

Wikipedia article (for further details of the stimuli see [1,14]). This ensured link presence was counterbalanced across sentences and participants. The manipulations of Word Type and Word Frequency were not shown to have any effect on importance ratings (all *p*'s >.10). As such, they were not considered within the current analyses. Please see Fitzsimmons et al. for full details of the stimuli, including the selection and implementation of single word analysis.

**Procedure.** Ethical approval for Experiment 1 was applied for, peer-reviewed and granted by the University of Southampton Psychology Department Ethics Committee. Participants were given an information sheet and a verbal description of the experimental procedure and informed that they would be reading Wikipedia articles on the desktop computer screen. The participants were instructed to read through the entire Wikipedia article on the desktop screen. Having completed reading the passage in its entirely, participants then looked at the laptop screen, in order to rate the importance of each individual sentence in sequence from that article. They were instructed to rate each sentence on how important it was to the general meaning of the article as a whole and respond using the buttons 1 (Not important)– 5 (Very important) on the laptop keyboard. Once all sentences had been rated, the next trial would appear. The experiment was self-paced and lasted approximately 60 minutes.

## Results

**Analyses.** We ran linear mixed models (LMMs) using the lme4 package in R [37] to explore the differences between importance ratings. Across analyses, Passage Type (Linked vs. Unlinked), Number of Links, Length of Sentence and Position on page (all three of which were centred) were treated as Fixed Factors. Participants and items were included as random effects variables. Across analyses, a maximal random model was initially specified for the random factors [38]. If a model did not converge, the random effect structure was pruned first by removing the interactions between the slopes, then correlations in the random structure and finally by successively removing the slopes for the random effects explaining the least variance until the maximal converging model was identified. Model comparisons were carried out to investigate whether the interactions added to the fit of the statistical models and the most parsimonious model will be reported. For all analyses, successive differences contrasts were used to explore fixed factor effects, such that the intercept corresponds to the grand mean and the fixed factor estimate for a categorical factor can be interpreted as the difference between the two conditions.

**Effect of hyperlinks.** To explore the effect of hyperlinks, we compared importance ratings from participants who viewed an entirely unlinked text versus those who saw the text with links. As such, only Passage Type (i.e. the between subjects variable of whether participants were completing the experimental task on linked or unlinked passages), Length of Sentence in characters and Position on the Page were included as fixed factors in this analysis.

We found a two-way interaction between Length of Sentence x Position on Page, which in turn was qualified by a higher-order three-way interaction between Passage Type x Length of Sentence x Position on Page (Fig 1). The two-way interaction between Length of Sentence x Position on Page exhibited that sentences at the top of the page are rated as more important than sentences at the bottom, but this effect was stronger for longer sentences. This two-way interaction was qualified by a three-way interaction that also includes whether the passage included links or not. Fig 1 shows that the two-way interaction is stronger and more pronounced (sharper slope) for the passages that contains links compared to the passages that do not contain links.

There was a main effect of Passage Type where sentences rated by those in the Unlinked group were rated higher than those rated by the Linked group (Mean for Unlinked = 3.23,

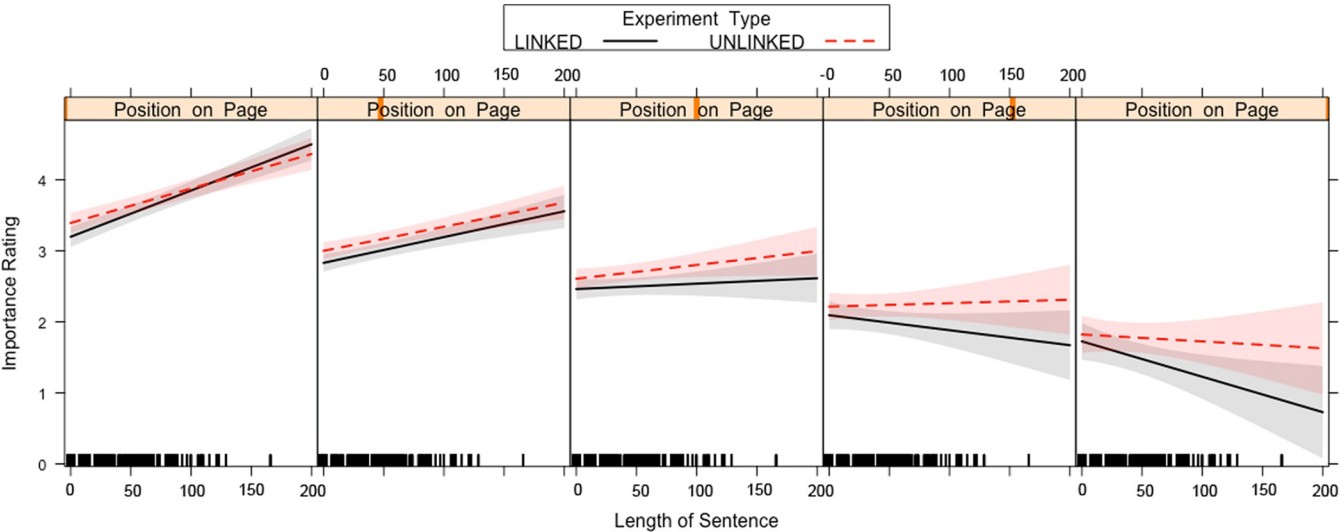

**Fig 1. Passage Type x Length of Sentence x Position on Page interaction for the importance ratings of the unlinked sentences in Experiment 1.** Each box on the graph represents the different positions a sentence can sit on the page, from top to bottom (left to right on the graph). A 95-percent confidence interval (the grey shaded region) is drawn around the estimated effect.

$SD$ = 1.23; mean for Linked = 3.10, $SD$ = 1.30; see Table 1 for the LMM analysis). This suggests that when an unlinked sentence is presented with other sentences that have links, the unlinked sentence is rated lower compared to when it is featured in a context without any hyperlinks. There was also a main effect of Length of Sentence where, regardless of whether the text was read with links or without links, the longer sentences are rated higher in importance than shorter sentences. A main effect of Position on Page was also present where, regardless of whether the text was read with links or without links, sentences closer to the top of the page are rated higher.

**Effect of linked vs. unlinked sentences.**  Subsequently, we explored whether the presence or absence of links affected the importance rating within the text that contained links. Data from the unlinked group were excluded from this analysis, and all subsequent analyses of Experiment 1. Whether the Sentence Contained Links, Length of Sentence in characters and Position on the Page were included as fixed factors. Model comparisons revealed a model

**Table 1. Fixed effects estimates, standard error and *t* value for LMM for Experiment 1 comparing the unlinked sentences in the linked and unlinked passages.**

|  | Estimate | Std. Error | *t* value |
|---|---|---|---|
| Intercept | **3.09** | **0.05** | **61.61** |
| Passage Type | **0.16** | **0.03** | **4.54** |
| Length of Sentence | $\mathbf{3.63 \times 10^{-3}}$ | $\mathbf{7.00 \times 10^{-4}}$ | **6.69** |
| Position on Page | **-0.24** | **0.01** | **-19.80** |
| Passage Type x Length of Sentence | $2.36 \times 10^{-4}$ | $9.238 \times 10^{-4}$ | -0.26 |
| Passage Type x Position on Page | 0.02 | 0.02 | 1.23 |
| Length of Sentence x Position on Page | $\mathbf{1.44 \times 10^{-3}}$ | $\mathbf{2.64 \times 10^{-4}}$ | **-5.47** |
| Passage Type x Length of Sentence x Position on Page | $\mathbf{7.08 \times 10^{-4}}$ | $\mathbf{3.50 \times 10^{-4}}$ | **2.02** |

Note: Random structure for model: (1|Participants) + (0+Position on Page|Participants) + (1|Items). Bold indicates | t| > 1.96.

containing the two-way interaction between whether the sentence contained links and length of sentences did not significantly add to the fit of the data compared to a model without and neither did the three way interaction. Subsequently, these two interactions were excluded from our analyses. The model is presented in Table 2.

We found a two-way interaction between Length of Sentence and Position on Page and another two-way interaction between Sentence Contains Links and Position on Page. The two-way interaction between Length of Sentence and Position on Page (Fig 2) displayed the same interaction reported for the unlinked sentences in the previous analysis. For sentences at the top of the page, long sentences were rated as more important than short sentences, whereas for sentences at the end of the page, long sentences were rated as less important than short sentences.

The two-way interaction between Sentence Contains Links and Position on Page can be observed in Fig 3. Sentences at the bottom of the page are ranked as lower in importance, and even more so when there are no hyperlinks present in that sentence.

There was also a main effect of whether the Sentence Contains Links with the sentence with links being rated as more important (average rating of sentence, with links = 3.28, *SD* = 1.25; without links: 3.10, *SD* = 1.30). There was also a main effect of Length of Sentence with longer sentences being rated higher and a main effect of Position on Page where sentences at the top of the page were rated higher.

**Effect of number of hyperlinks.** We also analysed the effect of the number of links in a sentence on importance ratings, in the texts that feature links. The LMM for this analysis did not include ratings from the sentences that did not feature any links. The Number of Links (a continuous factor, which was centred), Length of Sentence in characters and Position on the Page were all included as fixed factors.

There was a two-way interaction between Length of Sentence and Position on Page and a two-way interaction between Number of Links and Position on Page. Both interactions, however, were qualified by a three-way interaction between Number of Links and Length of Sentence and Position on Page. The two-way interaction between Length of Sentence and Position on Page (Fig 4) was the same in the previous two analyses and indicated that at the top of the page long sentences were rated as more important than short sentences, whereas for sentences at the end of the page long sentences were rated as less important compared to short sentences. The three-way interaction with Number of Links (Fig 5) qualifies this interaction in that this two-way interaction is only present for sentences that contain a low number of links and was not present for sentences that contain a high number of links.

The two-way interaction between Number of Links and Position on Page indicated that at the top of the page (Fig 5), sentences with more links were rated as being of more importance

**Table 2. Fixed effects estimates, standard error and *t* value for LMM model for Experiment 1 comparing sentences with and without links.**

| | Estimate | Std. Error | *t* value |
|---|---|---|---|
| Intercept | 3.11 | 0.09 | 35.49 |
| Sentence contains Links | 0.28 | 0.03 | 8.94 |
| Length of Sentence | $4.38 \times 10^{-3}$ | $4.44 \times 10^{-4}$ | 9.86 |
| Position on Page | -0.24 | 0.01 | -40.15 |
| Length of Sentence x Position on Page | $1.49 \times 10^{-3}$ | $1.57 \times 10^{-4}$ | -9.48 |
| Sentence contains Links x Position on Page | $3.61 \times 10^{-2}$ | 0.01 | 3.11 |

Note: Random structure for model: (1 |Participants) + (1|Items). Bold indicates |t| > 1.96.

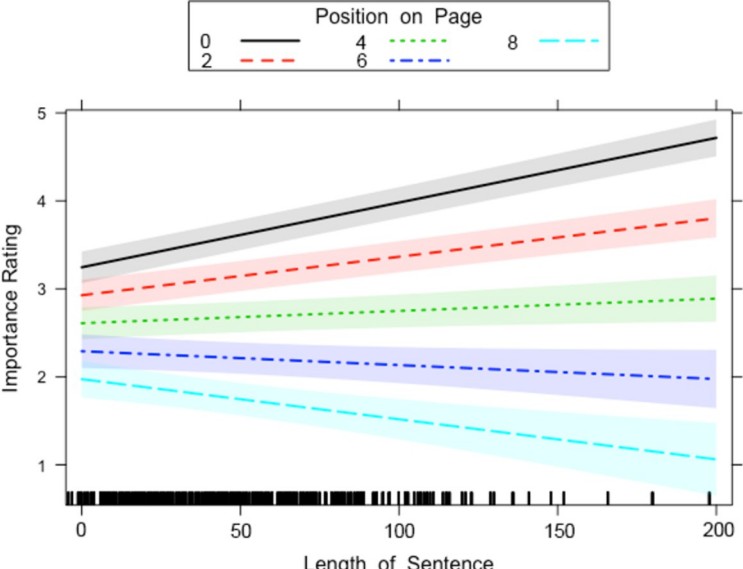

**Fig 2. Length of Sentence x Position on Page interaction for the linked passages in Experiment 1.** The lines on the graph represent the different positions a sentence can sit on the page. A 95-percent confidence interval (the grey shaded region) is drawn around the estimated effect.

than sentences with fewer links. Conversely, at the bottom of the page, sentences with fewer links were rated as being of more importance than sentences with more links. However, this is again qualified by the three-way interaction that also includes the Length of the Sentences (Fig 6), which indicated that the two-way interaction only occurred for short sentences and was absent in the case of long sentences. For long sentences, the sentences with a high number of links were rated as more important, regardless of the position on the screen.

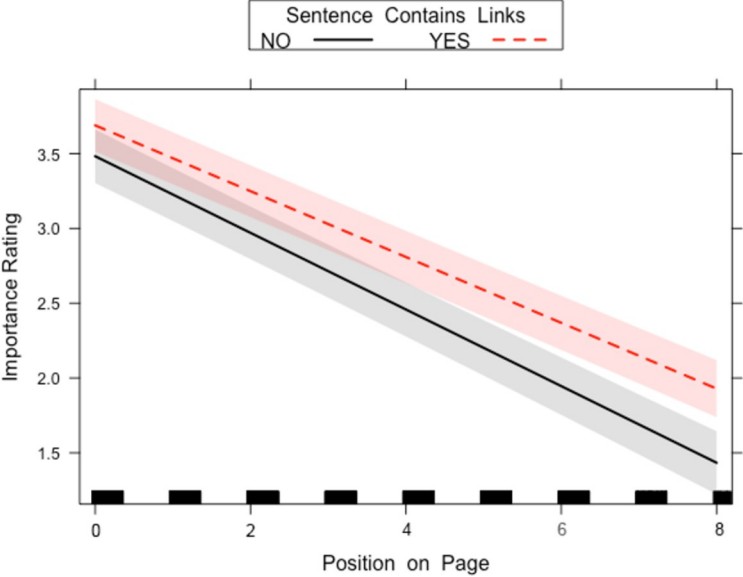

**Fig 3. Sentence Contains Links x Position on Page interaction for the linked passages in Experiment 1.** The lines on the graph whether the sentence contains links (dashed line) or not (solid line). A 95-percent confidence interval (the grey shaded region) is drawn around the estimated effect.

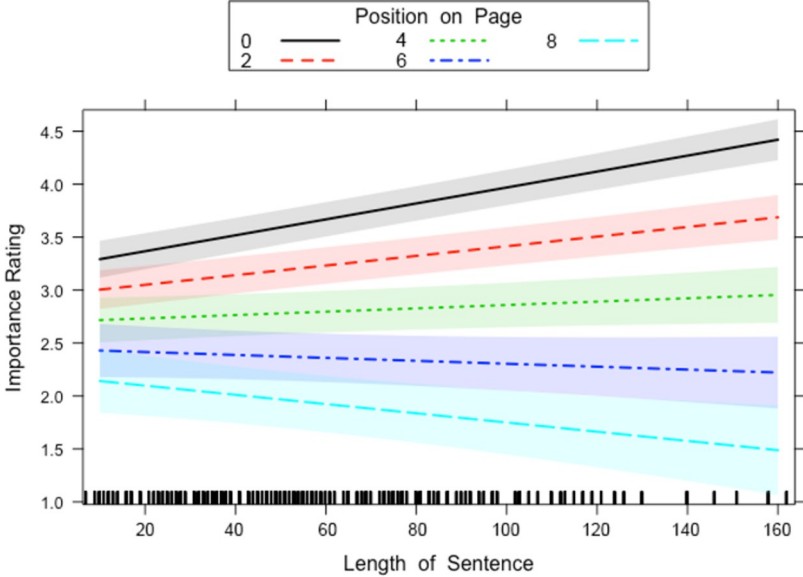

**Fig 4. Length of Sentence x Position on Page interaction for all sentences that contain links in Experiment 1.** The lines on the graph represent the different positions a sentence can sit on the page. A 95-percent confidence interval (the grey shaded region) is drawn around the estimated effect.

There was a main effect of Number of Links, whereby sentences with more links were rated higher for importance than fewer links. There was also a main effect of Length of Sentence, where longer sentences were rated higher than shorter sentences. There was also an effect of Position on Page, where sentences closer to the top of the page were rated as being of higher

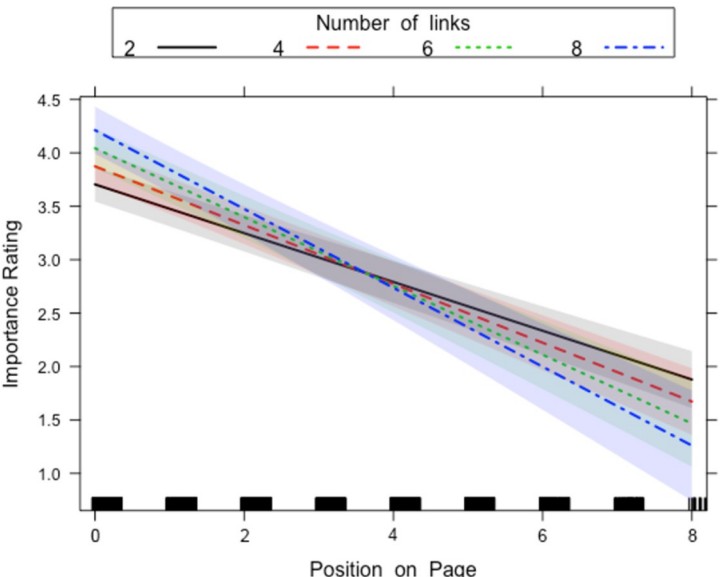

**Fig 5. Length of Sentence x Position on Page x Number of Links interaction for all sentences that contain links in Experiment 1.** The lines on the graph represent the number of links a sentence contains. Each box on the graph represents the different lengths of sentence from shortest to longest (left to right on the graph). A 95-percent confidence interval (the grey shaded region) is drawn around the estimated effect.

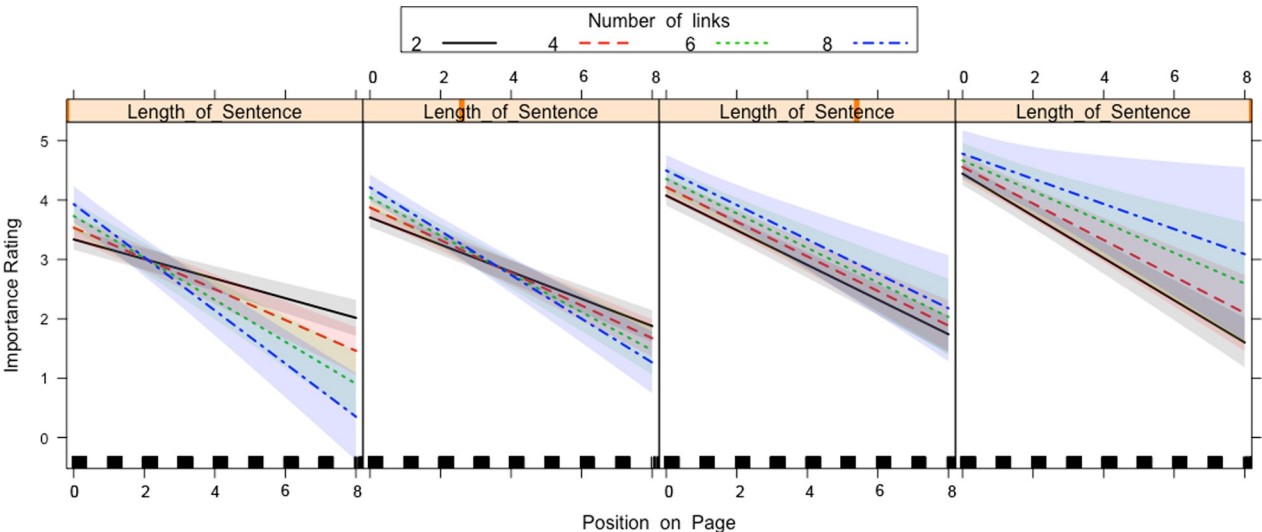

**Fig 6. Number of Links x Position on Page x Sentence Length interaction for all sentences that contain links in Experiment 1.** The lines on the graph represent the different positions a sentence can sit on the page. A 95-percent confidence interval (the grey shaded region) is drawn around the estimated effect.

importance than sentences closer to the bottom of the page The models are presented in Table 3.

## Discussion

Experiment 1 demonstrated that there are a number of factors that people use to rate the importance of a sentence. Our between groups analysis of the unlinked and linked tasks showed that when a sentence does not contain links but is in a hypertext environment with other sentences that do contain links, it is rated as being of lower importance. This suggests sentences that contain links are taking importance away from the sentences without links. This occurred regardless of the content of that sentence, as we were directly comparing sentences that were visually identical, the only difference between the unlinked sentences being the presence or absence of hyperlinks in the surrounding sentences. Overall, longer sentences were rated more important. Sentences at the top of the page were usually rated as more

**Table 3. Fixed effects estimates, standard error and *t* value for LMM model for Experiment 1 comparing the number of links in the sentences when they contain links.**

|  | Estimate | Std. Error | *t* value |
|---|---|---|---|
| Intercept | **3.17** | **0.09** | **35.09** |
| Number of Links | **0.04** | **0.01** | **2.73** |
| Length of Sentence | $3.53 \times 10^{-3}$ | $8.71 \times 10^{-4}$ | **4.06** |
| Position on Page | **-0.18** | **0.01** | **-12.29** |
| Number of Links x Length of Sentence | $6.60 \times 10^{-4}$ | $3.68 \times 10^{-4}$ | 1.79 |
| Number of Links x Position on Page | **-0.02** | $4.68 \times 10^{-3}$ | **-4.99** |
| Length of Sentence x Position on Page | $2.21 \times 10^{-3}$ | $3.05 \times 10^{-4}$ | **-7.26** |
| Number of Links x Length of Sentence x Position on Page | $4.73 \times 10^{-4}$ | $1.23 \times 10^{-4}$ | **3.87** |

Note: Random structure for model: (1+Position on Page| Participants) + (1|Items). Bold indicates |t| > 1.96.

important and adding hyperlinks to a sentence increased the perceived importance of that sentence. But there were also a number of interactions across these findings.

When comparing sentences that have links versus those that do not have links (but in an environment that always features some links), we found that sentences with links were rated for higher importance. Furthermore, we found an interaction of position on page and links whereby this effect was more apparent towards the bottom of the page. This could be because having a link present serves as a boost to importance and when a sentence near the bottom of the page would be rated quite low, if it contains a link it gets a boost of importance. This has less of an impact at the top of the page, where sentences were shown to generally be rated as very important. In terms of the effect of the number of links, we found that short sentences with more links were rated higher when placed at the top of page. The interaction between number of links and position on page showed, however, that this effect was reduced as the reader moves down the page. However, long sentences with many links were rated higher, no matter the position of the sentence on the page. This suggests that links do make a sentence more important, especially for long sentences.

Both of these findings align with previous literature noting the importance of the hyperlink in reading hypertext. Signalling research has shown that signals can help highlight the important sections of text [39], aid the memory of important sections [19–22], and increase the speed of lexical processing [23]. Furthermore, reading research has shown that hypertext environments lead readers to relying on links to extract key information from text [1]. Our findings support the notion that hyperlinks are used to signal important information. Important sentences containing one or more hyperlinks can be easily recognised due to the saliency of the coloured words and they can prove a very useful typographical cue for the readers. As such, readers judge sentences with more links as being of more importance to the understanding of the passage (in this case a webpage).

Throughout our analyses Position on Page seems to also have a key impact on the rating of sentences, and this was seen through its influence on the effect of other factors. Sentence length interacted with Position on Page in all our analyses when links were present, such that at the top of the page, a long sentence is rated as more important than a short sentence, whereas at the end of the page a short sentence is rated as more important than a long sentence. This could suggest that information at the top of the page is generally considered more important, and longer sentences here are perceived as more important as they contain more potentially important information, by virtue of their length. This could be considered similar to the information bias [30], whereby readers judge the presentation of more information as being more useful for understanding the text, just because more is presented. It must, however, be noted that this interaction was qualified by a higher order interaction with number of links. We found that for long sentences, those sentences with a large number of hyperlinks will always be rated the most important regardless of their position on the screen. This result supports the overarching importance of hyperlinks when reading on the Web. It also provides behavioural evidence to suggest readers seem to provide offline ratings of importance of information on the Web in a manner similar to the "inverted pyramid" structure of Webpages [31], even when we did not explicitly structure the semantic information in this manner (i.e. the most important information is expected to appear at the top of the page).

In sum, Experiment 1 provided evidence that the number of links is utilised by the reader to estimate the importance of a sentence when reading on the Web. Other factors such as the sentence length and position of the sentence on the page all have an impact on the importance rating of the sentence and the reader can use these factors to assume the importance of individual sentences. While this provides interesting evidence for how readers form offline interpretations after they have a webpage, it does not tell us how this affects online reading behaviour.

Experiment 2 utilised eye movement methodology, in order to investigate how importance, and other physical variables of a webpage, affect the allocation of attention during reading.

## Experiment 2

Experiment 2 focuses on how perceived importance and task effects (skim reading versus reading for comprehension) affects how individuals sample text on the Web and extract information from it. Experiment 1 found hyperlinks to be a key determinant of which parts of the text are perceived to contain important information and it could be predicted, therefore, that readers will judge the presence of links as an essential indicator for gaining a desired amount of comprehension.

With the large amount of information online, it can be safely assumed that skim reading is a common behaviour [2,6], which has been found to lead readers to use links to anchor their attention when reading on the Web [1]. As such, links are a high saliency signal and crucial for readers engaged in skim reading. Skim reading is an efficient way of gaining as much information as possible in the shortest amount of time, attempting to sacrifice as little comprehension as possible. When used to highlight important information [10], the reader can use links to efficiently identify important information in the text and move through the text faster, ignoring unimportant information and using their time to instead focus on the text flagged as important through this signalling. It is also possible that readers are using additional signals, such as those identified in Experiment 1, specifically sentence length, and position on page. If this were the case, we would expect readers to fixate sentences seen as more important (as a function of position on page, number of links and sentence length) for longer than those judged as being less important.

Whereas participants provided offline ratings in Experiment 1, Experiment 2 explores the differences between reading for comprehension and skim reading on reading behaviour by recording the participants' eye movements. We also employed the task manipulation from Fitzsimmons, Jayes, Weal and Drieghe [1], whereby participants were either instructed to read for comprehension or asked to skim read passages of text that resemble a Wikipedia page. Between each page of text, the participant was asked comprehension questions which were either related to the previously rated important or unimportant sentences in the text. The ratings from Experiment 1 were used to examine the impact of the perceived importance that was given to each sentence. This provided a highly controlled stimuli set, while our methodology provided a naturalistic, moment-by-moment indication of the processing of text during normal reading [12,13,17,40]. Furthermore, we did not employ clicking of links, as in Experiment 2 of the previous study [1], as this would have come at great cost for the number of importance ratings required. By allowing participants to click on links and thereby navigate a selection of web pages, the number of observations per target word would have fluctuated substantially between subjects. By employing our paradigm of using the same 40 Wikipedia passages, we were able to maintain experimental control over our stimuli, and the ratings of these stimuli as a result.

From previous research, we predicted that readers would read faster when asked to skim read, but would have reduced comprehension [1,26]. Whereas these previous studies have primarily focussed on single word target analyses (see [1]), we employed more global, sentence-based eye movement measures in the current study, as well as comprehension accuracy as an index of how well participants understood the text. We predicted shorter sentence reading times and more word skipping in the skim reading condition. In terms of importance, we predicted that sentences with higher importance ratings from Experiment 1 would exhibit longer sentence reading times, in comparison to the unimportant sentences. This would reflect the

readers prioritising information that they consider to be important during online language comprehension. Furthermore, from Experiment 1, we know that aspects of importance, when considered offline, relate to number of links, position on page and sentence length. If these are also used to consider importance online during reading, sentences previously rated as important should receive more time being processed by readers.

In addition, we also analysed reading times on the last word of each sentence to measure ease of text integration, known as wrap up reading times. Previous research has shown that wrap up measures are reflective of processing times related to the integration of text within and across sentences [41,42]. Specifically, wrap up processing has been shown to be positively associated with the variables of information density [43]), processing capacity [44] and literacy development [45]. Difficulty with the integration of text is associated with less skipping of the last word of a sentence and more time spent fixating that word. It is also possible that assessing sentence importance is an aspect of sentence processing, where the relative worth of the sentence just read is assessed once it has been read in totality. We included wrap up reading time analyses in order to explore this possibility.

We also predicted an interaction between task type and importance, where the effect of importance would be increased during skim reading. We predicted this as readers would need to rely more on typographical and physical markers over semantic content, in order to process the text as efficiently as possible, to allow skim reading. As links are salient in the text, they could easily be used as an efficient strategy for selecting the important sections of text, as could sentence length and position on page. As such, we predicted that when skim reading the reader may use these markers to judge where the important information lies in the page. Readers should therefore, compared to reading for comprehension, spend more time on sentences rated as more important when skim reading, to satisfy the need to read the passage quickly while still extracting the most important information.

As for whether there would be a difference in the comprehension of important and unimportant information, previous research suggests that during skim reading some comprehension may be lost [24–27]. However, this loss in comprehension is not consistent, there appears to be a difference between information regarded as important or unimportant. The important information does not receive the same loss of comprehension that is observed for the unimportant information [7,27,28]. If skim reading is an efficient strategy to read through text the fastest way possible while minimising comprehension loss, then we would expect that the skim readers will perform more poorly on comprehension question about the unimportant information. However, if there were a general reduction in comprehension across both important and unimportant sentences, then there would be a straightforward speed-accuracy trade-off for skim readers.

## Method

**Participants.** Thirty-two native English speakers (2 males, 30 female) with an average age of 20.00 years participated in exchange for course credits or payment (£9) and were members of the University of Southampton community. All had normal or corrected-to-normal vision and no known reading disabilities. None of the participants took part in Experiment 1. Post hoc calculations of power were conducted given the current sample size using the simr package in R [35] and consistently returned an estimated power above 80% with the significance level of $\alpha$ = .05 (as suggested by Cohen [36]). Across the analyses for Experiment 2, power values ranged between 84–100% for main effects (effect sizes ranged between .03-.44). Ethics approval was applied for, peer-reviewed and granted by the University of Southampton Psychology

Department Ethics Committee. Participants gave written consent before participating in the studies.

**Apparatus.** Eye movements were measured with an SR-Research Eyelink 1000 eye tracker operating at 1000 Hz (1 sample every millisecond). Participants viewed the stimuli binocularly, but only the right eye was tracked. Words were presented in 14pt mono-spaced Courier font. The participant's eye was 73 cm from the display; at this distance three characters equalled about 1˚ of visual angle.

**Stimuli and design.** Stimuli used were identical to those in Experiment 1. The study employed the continuous independent variable of importance (i.e. the average importance scores for each sentence from Experiment 1) and the independent variable of Task Type (Comprehension, Skimming). Both variables were within subjects, with participants reading all sentences, with the task of skim reading and reading for comprehension counterbalanced across the passages.

**Procedure.** Ethical approval for Experiment 2 was applied for, peer-reviewed and granted by the University of Southampton Psychology Department Ethics Committee. Participants were given an information sheet and a verbal description of the experimental procedure and informed that they would be reading passages on a monitor while their eyes were being tracked. The text on the screen gave the instructions to read either for comprehension or to skim read. This was blocked such that the first twenty stimuli were to be read for comprehension and the second twenty to be skim read.

When the skim reading portion of the experiment began the participants were instructed to 'skim read as you would naturally, as if you are reading a large textbook that you need to read quickly'. Participants were told there was no time limit, and they simply had to skim read naturally. As in Fitzsimmons et al. [1], we did not counterbalance the order of Task Type because the comprehension reading blocks might have been influenced by first having to skim read. Participants were not told they were going to be skim reading until just before that half of the experiment was due to begin, so as not to influence the first part of the experiment which was to be read for comprehension. If participants are first asked to skim read, it may become difficult to slow down and read "normally" afterwards and this would affect our data as we would not observe normal reading behaviour.

The participants were informed that they were to respond to comprehension questions presented after each trial. The participants' head was stabilised in a head/chin rest to reduce head movements that could adversely affect the quality of the calibration of the eye tracker. A 9-point calibration procedure preceded the experimental trials. A maximum error of .5 degrees was permitted. At the beginning of each trial the participant had to look at a fixation point on the screen. When the eye tracker registered a stable fixation on the fixation point, the sentence was displayed ensuring that the first fixation fell at the beginning of the text. When participants finished reading, they confirmed they had finished by pressing a button on the response box in front of them. After each trial, four comprehension questions were presented to the participants, one at a time. Two of the questions were related to sentences within the passage rated as the most important in the hyperlinked portion of Experiment 1. The other two questions were related to the sentences rated as the least important in Experiment 1. Each comprehension question required a yes or no response and tested text-based comprehension. For example, following a passage on American Football, participants were asked *Is American football played between two teams of thirteen*? The comprehension questions were presented to ensure the participants were comprehending the text displayed to them and also to measure the level of comprehension across both the sentences rated as important and unimportant. Participants responded to the questions by pressing the appropriate button on a response box. After the

questions the next trial would appear. The experiment lasted approximately 60 minutes, with a 15-minute break in between task type blocks.

## Results

As in Fitzsimmons et al., 2019, eye movement data was processed using EyeLink's Data Viewer software. Areas of interest were calculated according to the X and Y axis coordinates of the sentences used, with fixations within these areas combined according to the nature of the measure (see 'Global Measures'). Short, contiguous fixations were corrected using an automatic procedure; fixations under 80 milliseconds were incorporated into larger fixations within one character, and both short fixations under 40 milliseconds and more than three characters from another fixation, and long fixations over 800 milliseconds, were deleted. In addition, we eliminated trials in which there was track loss or participants appeared not to have completed reading the passage. Prior to data analysis, data for each eye movement measure more than 2.5 standard deviations from each participant's condition mean were removed (affecting <1% of data).

**Global measures.**    Five eye movement measures were calculated, two were based on fixation times for each sentence and three based on wrap up reading times. First pass sentence reading time includes all fixations in the first pass reading of the sentence (i.e. fixations initially made on a sentence before fixating another). We also calculated total reading time, as 23.72% of fixations were part of rereading behaviour. Total reading time consisted of all fixations on the sentence including all rereading. A number of wrap up measures were also calculated for the final word of target sentences (which were fixated 70.44% of the time). Wrap up times were used as they are considered to be reflective of processing times related to the integration of text within and across sentences [41,42]. Wrap up skipping probability is the probability that the last word did not receive a direct fixation during first pass reading. First pass wrap up reading time is the summed duration of fixations from the first fixation on the last word until readers made a saccade away from that word. Wrap up total time consisted of all fixations on the last word, including all rereading (in 19.78% of trials the last word was fixated more than once).

**Analyses.**    We ran LMMs using the lme4 package in R [version 1.1–26, 23] to explore the impact of two independent variables which were included as fixed factors: Task Type (Comprehension, Skimming) and Importance Rating (based on the scores from Experiment 1, 1–5 and centred). Binominal models were used for the wrap up skipping probability measure. An interaction was included between the fixed factors unless model comparisons proved that the model was a better fix without the interaction term. Participants and items were included as random effects. Across analyses, a maximal random model was initially specified for the random factors [38] with the same pruning procedure as in Experiment 1. All reading time measures were log transformed, in order to normalise skewed data. All means and standard deviations are showed in Table 4 and all fixed effects estimates are shown in Table 5.

We found an interaction between Task Type and Importance Rating, for the measure of total reading time. While longer total reading times were observed for more important

**Table 4. Means of eye movement measures for Experiment 2.** Standard Deviation in parentheses.

| Task Type | First Pass Sentence Reading Time (ms) | Total Sentence Reading Time (ms) | Wrap Up Skipping Probability (%) | First Pass Wrap Up Reading Time (ms) | Wrap Up Total Reading Time (ms) |
|---|---|---|---|---|---|
| Comprehension | 2631 (1886) | 3558 (1914) | 53.93 (49.85) | 246 (113) | 290 (152) |
| Skimming | 1493 (993) | 1825 (983) | 62.01 (48.54) | 207 (70) | 221 (86) |

**Table 5. Fixed effect estimates for global eye movement measures in Experiment 2.**

| | First Pass Sentence Reading Time (ms) | | | Total Sentence Reading Time (ms) | | | First Pass Wrap Up Reading Time (ms) | | |
|---|---|---|---|---|---|---|---|---|---|
| | Estimate | Std. Error | t value | Estimate | Std. Error | t value | Estimate | Std. Error | t value |
| Intercept | **6.92** | **0.08** | **86.86** | **7.12** | **0.08** | **89.13** | **5.33** | **0.02** | **252.56** |
| Task Type | **-0.44** | **0.07** | **-5.98** | **-0.63** | **0.04** | **-15.34** | **0.16** | **0.01** | **13.87** |
| Importance Rating | **0.10** | **0.07** | **4.83** | **0.18** | **0.02** | **8.59** | **0.03** | **0.01** | **3.01** |
| Task Type x Importance Rating | -0.01 | 0.02 | -0.22 | **-0.02** | **0.01** | **-2.35** | **0.03** | **0.01** | **2.07** |
| | Wrap Up Skipping Probability | | | Wrap Up Total Reading Time (ms) | | | | | |
| | Estimate | Std. Error | z value | Estimate | Std. Error | t value | | | |
| Intercept | **0.83** | **0.20** | **4.19** | **5.43** | **0.02** | **223.14** | | | |
| Task Type | **-0.36** | **0.17** | **-2.05** | **0.25** | **0.02** | **11.25** | | | |
| Importance Rating | **-0.13** | **0.04** | **-2.92** | **0.05** | **0.01** | **3.82** | | | |
| Task Type x Importance Rating | | | | **0.06** | **0.01** | **3.74** | | | |

Note: Random structure for first pass sentence reading time, total sentence reading time, wrap up skipping probability: (1 + Task Type | Participants) + (1|Items), first pass wrap up reading time: (1 + Importance |Participants) + (1|Items), wrap up total reading time: (1 + Task Type*Importance | Participants) + (1|Items). Bold indicates |t| > 1.96.

sentences, this effect was more pronounced (i.e. steeper slope) in the reading for comprehension condition, compared to the skim reading condition (see Fig 7). For first pass sentence reading time and total sentence reading time, we found a main effect of Task Type, where there were longer first pass and total reading times on a sentence when it was read for comprehension compared to skim reading. This replicates previous findings [1] but within global, sentence level measures of reading. We also found a main effect of the Importance Ratings, where the higher the rating, the longer the first pass and total reading times. This means that readers spend longer on sentences rated as more important than those rated as less important, indicating that Task Type and Importance affect both first pass and rereading. The lack of an interaction between Task Type and Importance in first pass sentence reading time could suggest

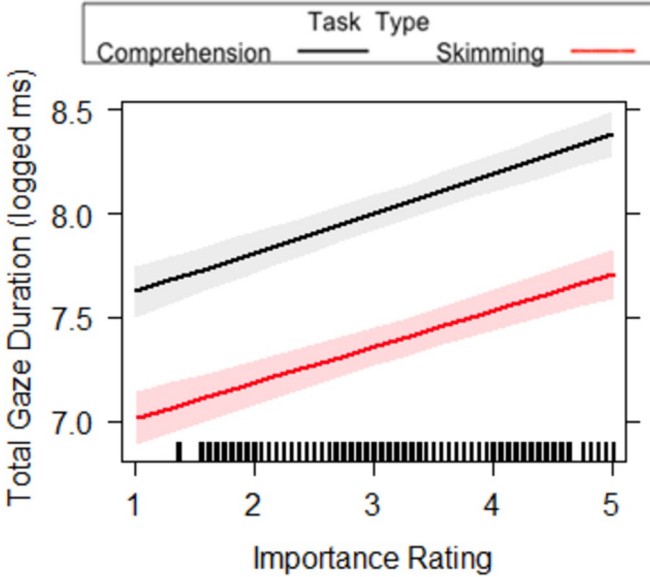

**Fig 7. Task Type x Importance interaction for total sentence reading time in Experiment 2.** A 95-percent confidence interval (the shaded region) is drawn around the estimated effect.

**Table 6. Fixed effect estimates for first pass and total sentence reading time per character in Experiment 2.**

| | First Pass Sentence Reading Time per Character | | | Total Sentence Reading Time per Character | | |
|---|---|---|---|---|---|---|
| | Estimate | Std. Error | *t* value | Estimate | Std. Error | *t* value |
| **Intercept** | **2.68** | **0.04** | **73.00** | **3.31** | **0.05** | **68.51** |
| **Task Type** | **0.46** | **0.04** | **11.39** | **-0.63** | **0.04** | **-15.40** |
| **Importance Rating** | **-0.04** | **0.02** | **-2.36** | **0.05** | **0.01** | **5.42** |
| Importance Rating * Task Type | 0.00 | 0.02 | 0.21 | -0.02 | 0.01 | -2.35 |

Note. indicates |t| > 1.96. Random structure for both first pass and total sentence reading time models: (1+Skimming|Participants) + (1|Items). Both measures were log transformed.

importance was not affected differently by whether the reader was skim reading or reading for comprehension during first pass reading. The presence of an interaction for total reading time does, however, suggest important sentences are reread more, but only when reading for comprehension, as opposed to skim reading where rereading is uncommon.

As it was shown in Experiment 1 that important sentences are longer, we ran additional analyses on global measures, more specifically on first pass sentence reading time per character and total reading time per character. This is essentially a measure of reading rate independent of sentence length (see Table 6). For first pass sentence reading time per character we replicated the main effects of Task Type and Importance Ratings and a lack of significant interaction between the two. Once again, we found there were longer first pass reading times per character when it was read for comprehension (*M* = 25.49, *SD* = 14.64) compared to skim reading (*M* = 14.59, *SD* = 8.00). We also found reading rates to be slower for sentences with higher importance ratings, compared to lower importance ratings. This suggests these effects are a result of our manipulations, rather than just a function of sentence length. For total sentence reading time per character, we replicated a main effect of Task Type, main effect of Importance Rating and a significant interaction between the two (see Fig 8). Once again, while

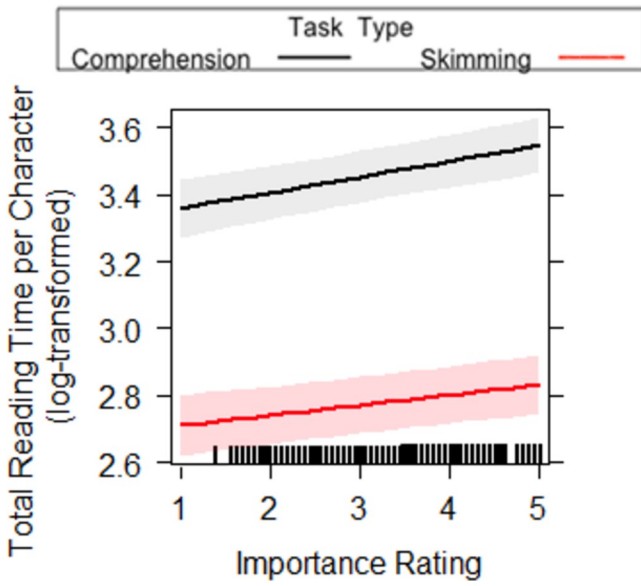

**Fig 8. Task Type x Importance interaction for total sentence reading time per character in Experiment 2.** A 95-percent confidence interval (the shaded region) is drawn around the estimated effect.

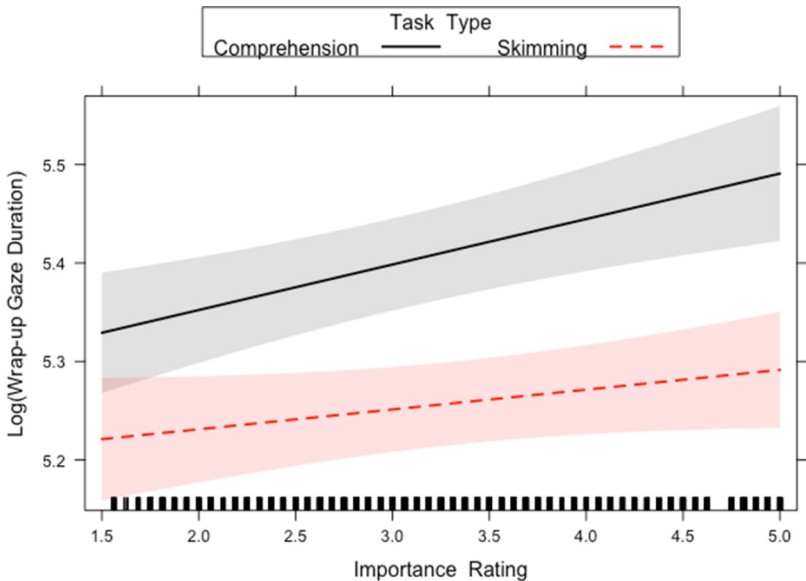

**Fig 9. Task Type x Importance interactions for first pass wrap up reading time in Experiment 2.** The lines on the graph represent the different tasks. A 95-percent confidence interval (the shaded region) is drawn around the estimated effect.

longer total reading times were observed for more important sentences, this effect was more pronounced (i.e. steeper slope) in the reading for comprehension condition, compared to the skim reading condition. Furthermore, the main effects showed there were longer first pass reading times per character when it was read for comprehension compared ($M = 34.17$, $SD = 13.48$) to skim reading ($M = 17.69$, $SD = 7.59$). We also found reading rates to be slower for sentences with higher importance ratings, compared to lower importance ratings.

Wrap up skipping had a main effect of Task Type and Importance Rating. Participants skipped the final word of a sentence more when skim reading compared to comprehension reading and they also skipped more when the sentences were rated as lower in importance. This suggests that the readers are trying to efficiently process the important and unimportant information, as they are showing increased skipping of the wrap up region of unimportant sentences compared to the important sentences.

Finally, first pass wrap up reading time and wrap up total time both showed a main effect of Task Type and Importance Rating. This was qualified by an interaction between Task Type and Importance Rating (see Figs 9 and 10). While longer first pass and total durations were observed for more important sentences, this effect was more pronounced (i.e. steeper slope) in the reading for comprehension condition, compared to the skim reading condition. This suggests a reduced effect of importance in skim reading behaviour. Furthermore, as wrap-up effects reflect higher level integrative processes, it suggests this level of processing is reduced in skimming conditions.

**Comprehension.** Analyses for comprehension scores were the same as in previous analyses, with the exception that Importance was treated as a categorical Fixed Factor (High Importance, Low Importance), with two questions asked about the two most important sentences from each passage and two about the least important (based on the ratings from Experiment 1). For the fixed factors successive differences contrasts were used such that the intercept corresponds to the grand mean and the fixed factor estimate for a categorical factor can be interpreted as the difference between the two conditions.

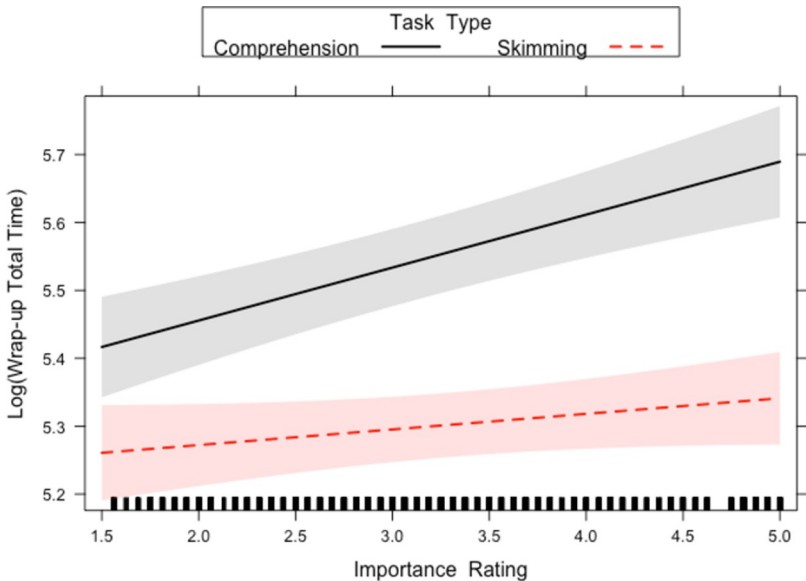

**Fig 10. Task Type x Importance interaction for wrap up total time in Experiment 2.** The lines on the graph represent the different tasks. A 95-percent confidence interval (the shaded region) is drawn around the estimated effect.

There was a main effect of Task Type where accuracy was significantly lower when the text was being skim read ($M = 0.86$, $SD = 0.13$) than when it was read for comprehension ($M = 0.91$, $SD = 0.08$), see Table 7 for LMMs). This replicated previous research suggesting that comprehension is impaired when skim reading [1,25–27,46]. There was no main effect of Importance Ratings (High–$M = 0.89$, $SD = 0.10$; Low–$M = 0.87$, $SD = 0.11$), or interaction between Importance Ratings and Task Type.

In addition, we used the same methodology for analysing the comprehension question results as Duggan and Payne [7] where Signal Detection Theory (SDT) measures were used to explore participants' comprehension of the text. SDT has been used in a wide range of visual cognition tasks [47]. SDT measures consider the proportion of trials where participants respond correctly (termed 'hits') as well as the proportion of trials where participants erroneously respond 'yes' (termed 'false alarms'). Duggan and Payne [7] used the SDT measure $d'$ to examine overall response accuracy, with higher values indicating better overall response accuracy. They also used the SDT measure $c$ to measure the response criterion or bias. Higher values of the response criterion indicate a tendency to respond 'no', suggesting that participants are 'biased' towards more conservative responses (i.e., they only respond 'yes' when there is a strong reason to do so). Lower values of the criterion indicate a tendency to respond 'yes', suggesting that participants are biased towards more liberal responses (i.e., they are willing to respond 'yes', even when there is only a weak reason to do so). We used these measures and examined them using a 2 (Importance: High Importance, Low Importance) x 2 (Task Type: Comprehension, Skimming) within-subjects ANOVA (see Table 8 for means). For $d'$ there was a main effect of Task Type ($F(1,31) = 10.38$, $p < .001$). The participants' comprehension of the text decreased when they were skim reading. There was also a marginal main effect of Importance ($F(1,31) = 3.97$, $p = 0.06$), which suggests that the participants were to a degree engaged in an adaptive strategy because they had improved accuracy for comprehension questions relating to the most important information. There was no significant interaction between Importance and Task Type ($F(1,31) = 0.31$, $p = 0.58$). When examining the bias ($c$) there were

**Table 7. Fixed effect estimates for comprehension question accuracy in Experiment 2.**

|  | Estimate | Std. Error | *t* value |
|---|---|---|---|
| **Intercept** | **0.91** | **0.01** | **68.54** |
| **Task Type** | **-0.04** | **0.02** | **-2.23** |
| Importance Rating | 0.01 | 0.02 | -0.34 |
| Task Type * Importance Rating | -0.02 | 0.03 | -0.63 |

*Note. indicates |t| > 1.96. Random structure for comprehension model: (1 |Participants).*

no significant differences between the measures (all *F*'s were smaller than 2.90, all *p*'s were larger than .10). This shows that there was no bias when responding to the comprehension questions, i.e. participants were no more likely to respond yes or no.

## Discussion

Experiment 2 demonstrated that skim reading has a pronounced influence on reading behaviour and how readers utilise perceived importance during online sentence comprehension. Firstly, we replicated the findings of Fitzsimmons et al. [1] and Just and Carpenter [26], in showing that skim reading leads to shorter fixations and increased word skipping, as indicated by our main effect of Task Type across most global sentence reading measures. Whereas Fitzsimmons et al. [1] focussed on single word measures, this study extends these findings for more global reading patterns, showing the effects of skimming are consistent on a sentence level. We extended these findings in the current paper by analysing the influence of perceived sentence importance.

In most of the global sentence reading measures, we found a main effect of Importance, where sentences with higher ratings of importance were fixated for longer. This means that the readers did spend longer on sentences rated as more important than those rated as less important. Furthermore, in first pass wrap up reading time and total reading time there was an interaction between task type and importance rating. Higher importance ratings resulted in longer first pass reading times and longer total time on the wrap up region. However, in skim reading the importance rating did not seem to have such a large impact as it does when read for comprehension. This suggests that when skim reading, the reader has less opportunity to establish what is less and more important and therefore cannot utilise importance as successfully as when reading for comprehension. This is interesting, as it could be considered that the use of the typographical cues that are related to importance could be assessed without lexical processing, as they are relatively low-level signals to the reader (hyperlinks, length of sentence and position on the screen). Even so, our results indicate that while readers use these cues when reading for comprehension, it appears readers do not to use this information to the same extent when skim reading. As such, it seems that more time and processing resources are required to establish importance of text, even when signalled topographically. This sentence-level result, interestingly, is partially contrary to previous research, at the individual word level, showing prioritisation of visually salient words during skim reading [1] as importance was partially explained by the use/number of hyperlinks (see General Discussion).

We also observed a decline in comprehension accuracy when the participants were skim reading, but we also found that they performed somewhat better on the comprehension questions regarding the sentences that were rated as more important. This effect was, however, only marginal, perhaps as a result of a ceiling effect of high overall accuracy. While we did find a comparatively lack of use of Importance to allocate attention when skim reading at a sentence level in the current research, it has been seen that readers do use links as anchors of

**Table 8. Behavioural results containing accuracy, sensitivity, and criterion in Experiment 2.** Standard deviation in parentheses.

| Task Type | Importance | Accuracy Percentage | *d'* | *c'* |
|---|---|---|---|---|
| Comprehension | High Importance | 91 (5) | 3.03 (0.67) | -0.29 (0.35) |
| | Low Importance | 90(5) | 2.63 (0.74) | -0.32 (0.41) |
| Skimming | High Importance | 87 (7) | 2.90 (0.67) | -0.22 (0.35) |
| | Low Importance | 84 (6) | 2.40 (0.64) | -0.40 (0.39) |

attention at the individual word level. Taken together with the result of improved comprehension for important sentence when skim reading, this could suggest that the participants were prioritising the more important information effectively. Given sentences with more links were rated as more important, and previous research has shown that skim reading leads to increased focus on links [1], it seems readers use links as signals through the text to anchor attention, leading to increased comprehension of those sentences. This supports previous research showing signalled content leads to improved memory [19–22] and retention of hyperlinked text [48].

## General discussion

Across Experiment 1 and 2, we investigated how perceived importance is influenced by properties of the materials being read and influences the reading of webpages. In Experiment 1, participants read Wikipedia pages and rated the importance of individual sentences, which were analysed according to the presence of links, the number of links, the position on the page and sentence length. We found longer sentences, sentences higher up the page and sentences with more links were rated as being of higher importance. Furthermore, we found a three-way interaction whereby the effects of sentence length were found to only occur at the top of the page and long sentences were actually of less importance at the bottom of the page, and this interaction occurred only for sentences with a low number of hyperlinks. In Experiment 2, we introduced an eye tracking methodology and the task manipulation of skim reading vs. reading for comprehension, to investigate how importance affects reading behaviour. We provide further evidence for the effects of skim reading, as well as the impact of perceived importance when reading. We observed that perceived importance was less influential during skim reading, with more important and less important sentences fixated for similar amounts when skim read. Despite our importance ratings not being based on semantic content, but rather based on relatively low level cues, (i.e. typographical information such as where the sentence was on the page, how long the sentence was, and how many hyperlinks it had), which should be fairly straightforward to extract from the text even in a limited time window, readers appear to not utilise them to assess importance during skim reading.

In Experiment 1, we found that typographical cues are critical in participants' ratings of importance, which subsequently affects reading behaviour on the Web in Experiment 2. These findings are consistent with signalling theory, such as SARA [49], a theoretical framework for signals. SARA considers text signals are expressions of an authors' intentions for the reader, such as providing headings to signal to the reader that the text is changing subject, and the author wishes to bring this to the attention of the reader to provide a more coherent reading experience for the reader. Research has typically focussed on the effects of signals on text processing, primarily exploring text headings as signals [50,51]. These findings primarily show other topographical signalling also supports text processing and improve memory of the text.

There are two key signal functions within SARA that are relevant with regard to hyperlinks: 1) 'emphasize a part of the text' and 2) to 'identify a function of a part of a text' [52]. Hyperlinks

both emphasize a part of the text through their physical properties and identify a key function of the text in their role of navigating readers to a new webpage. In Experiment 1, we found the number of links led to increased importance ratings, and reduced the effects of position on the page, i.e. the decrease in importance ratings for sentences lower down the page was reduced when sentences had more links. These findings underline the role of hyperlinks in text processing, as we could expect based on SARA's signal functions. We can clearly see how the physical properties of a hyperlink (colouring and underlining) signal to the reader the function of that part of text (i.e. they can be used to navigate to a new section of text). These signals subsequently alter a readers' perceptions of that sentence, and this was found for offline ratings of text and in online processing of text (where importance is directly related with number of links). Furthermore, this overrides other considerations of importance, particularly position on the page. Using SARA as a framework, it could be suggested that the multi-faceted signal functions of hyperlinks cause them to be considered as a more important typographical cue than others.

Further evidence for the increased signal function of hyperlinks is the notion that hyperlinks represent a different function to mere highlighting of text. Previous research has found differential effects of hyperlinks and highlighted words. While highlighted text has relatively little effect on reading, hyperlinks are re-read more often, but only when they are low frequency links [10]. As a result, it has been suggested that low frequent/difficult words can cause re-reading to assess the reasoning for why such a word would be highlighted as a link. The current research further suggests we should also consider this on a sentence level and whether links should be placed into less important sentences, potentially causing readers to misallocate importance as a result.

As a result of our considerations of typographical cues, however, our study is not primarily concerned with semantic and affective issues that affect perceived importance of text. Previously, studies have shown perceptions of the importance of text are affected by the degree to which mental imagery and affect are evoked by text [53] and as a function of subjective interest [54]. These factors have been shown to lead to increased engagement and subsequent memory for the text. Further research is required to investigate the degree to which typographical cues and reading task influence and possibly override more traditional, subjective variables of perceived text importance, and the effects on text processing. Furthermore, it must be considered that hyperlinks are not always used as signals, but also for the purpose of advertising and other commercial purposes. Links in Wikipedia are also based on explaining or describing an entity mentioned in the text [55], meaning many links will not be relevant to the immediate task of understanding the current webpage, but of tangential interest to a reader. Given the fact that hyperlinks seem to significantly affect readers perceptions of text importance, this adds to previous work suggesting education interventions may be required to foster an understanding how links are not always useful signals [3]. It has also been shown that disclosure and branding could be provided to alter perceptions of hypertext [56]. Taken together, it would seem increased awareness of the interaction to semantic and topographic cues may help improve digital reading strategies. Further research is also, therefore, required to investigate how 'good' digital readers' reading behaviour may differ from less efficient digital reading behaviour.

Further research also needs to consider the degree to which oversignalling could influence the use of signals. If a sentence has links that signal importance, but the semantic content of that sentence does not match this level of importance, the signals become an inefficient signal of importance. If a reader realises links are not a reliable signal of importance, it could cause them to be ignored and have less of an effect on text processing. This has been observed in studies of oversignalling, where inefficient signals have resulted in readers ignoring them [19].

The range of analyses provided show that importance affects reading behaviour at all stages of processing. Participants devote longer reading times to sentences of higher importance during first pass reading and also when we additionally included fixations after first pass reading (total sentence reading time). These effects were also replicated for wrap-up reading times both when we considered first pass reading time and when we included re-reading (total reading time). These latter results indicate that readers also spend more time integrating important sentences into their mental models of text, as this is the type of processing wrap up effects are most commonly associated with [41,57]. Wrap-up processing has previously also been shown to increase as a function of information density [43]. The present findings extend this finding to suggest wrap up processing also increases as a function of perceived sentence importance. Furthermore, this occurred when importance was decided by signalling variables, rather than informational content. We also found an effect of skim reading on wrap up reading times. When readers were engaged in skim reading, a reduction in wrap up reading time was also observed. In addition, readers also exhibit less marked increases in wrap up processing for important sentences when engaged in skim reading, compared to reading for comprehension. This suggests a lack of integrational processing is a factor in the compensatory strategic trade-off between time and comprehension previously observed for skim reading [1,58]. Specifically, readers seem to engage in less integrational processing during skim reading as part of this trade off.

The reduced influence of importance during skim reading in Experiment 2 adds to our growing understanding of how the nature of reading on the Web affects reading strategy. In Fitzsimmons et al. [14], we found hyperlinks had relatively little effect on reading behaviour (with the exception of low frequency hyperlinked words, which increased rereading). This was, however, only the case when reading static webpages for comprehension. In a follow up publication [1], we found hyperlinks do affect reading behaviour when skim reading (i.e. when reading in a more realistic Web environment). When skim reading static webpages, only linked words appeared to be fully lexically processed (evidenced by a lack of frequency effect for unlinked words), whereas both linked and unlinked words were fully lexically processed when reading for comprehension. When the task of navigation was subsequently introduced, linked words were the only words that appeared to be fully processed, regardless of whether skim reading or reading for comprehension. This suggested links were important signals for readers as they move through the text and were critical for skim reading. In the current study, we appear to find a reduced impact of links, as perceived importance had less of an effect on reading behaviour when skim reading, compared to reading for comprehension. Taken together, the findings of the current study and previous work [1] seems to suggest that while hyperlinks are of importance when making offline importance ratings, this is less the case during skim reading. It would appear that while hyperlinks are useful cues for navigating text [1], readers are less likely to be establishing importance during skim reading of static webpages. This highlights the difference between perceived importance and usefulness for navigation. Links are critical for the latter but appear to be less important for the former.

Further research is also needed to investigate how informational goals of the reader affect their perceptions of importance. While importance did not seem to affect reading behaviour, at a sentence level, when skim reading, further investigation is required to see whether this is modulated by other informational goals. This is of particular importance as the SARA model specifically notes the role of informational goals in the effectiveness of signalling [49], and previous research showing how tasks affect reading behaviour [59]. Our current results provide a baseline for how signals are used when reading Webpages for comprehension, or skim reading. Further research is required to investigate whether the interpretation and use of signals are affected by informational goals.

In terms of applications, the current research provides suggestions for the optimal presentation of information to readers. The current study found effects on offline importance ratings and online reading behaviour based on the topography of information presented. This means these relatively low-level informational cues of hyperlinks, sentence length and position on the screen can be used when considering website design and information presentation. Firstly, readers consider information presented at the top of the page more important. As such, it suggests the 'inverted pyramid' structure of information presentation [31] is an efficient method, or at least one readers have adapted and are sensitive to. Secondly, longer sentences are only considered to convey important information when they are presented towards the top of the page, suggesting longer sentences should be avoided if conveying important information towards the bottom of the page. Similarly, readers do not consider multiple hyperlinks to signal importance towards the bottom of the page. As such, excessive hyperlinking should be avoided in these sections of webpages (notwithstanding the issue of oversignalling discussed above [19]). This being said, it should also be considered that if text is highly likely to invite skim reading, it is unlikely that these considerations will influence the reader, due to the lack of effects of perceived importance on reading behaviour when engaged in this task. As such, the current study provides insight into optimal presentation of information for readers when reading hypertext.

In summary, we found that the perceived importance of sentences on webpages was influenced by the typographical cues of presence and number of hyperlinks, position on page and sentence length. By introducing these offline scores into our analysis of online reading behaviour, we found an influence of perceived importance on global and wrap up eye movement measures. We observed in readers a reduced influence of perceived importance during skim reading compared to reading for comprehension, suggesting these typographical cues are relied on less than may be expected when engaged in a reading strategy where there is a trade-off between comprehension and speed.

## Author Contributions

**Conceptualization:** Lewis T. Jayes, Gemma Fitzsimmons, Mark J. Weal, Johanna K. Kaakinen, Denis Drieghe.

**Data curation:** Lewis T. Jayes, Gemma Fitzsimmons, Mark J. Weal, Johanna K. Kaakinen, Denis Drieghe.

**Formal analysis:** Lewis T. Jayes, Gemma Fitzsimmons, Mark J. Weal, Johanna K. Kaakinen, Denis Drieghe.

**Funding acquisition:** Lewis T. Jayes, Gemma Fitzsimmons, Mark J. Weal, Johanna K. Kaakinen, Denis Drieghe.

**Investigation:** Lewis T. Jayes, Gemma Fitzsimmons, Mark J. Weal, Johanna K. Kaakinen, Denis Drieghe.

**Methodology:** Lewis T. Jayes, Gemma Fitzsimmons, Mark J. Weal, Johanna K. Kaakinen, Denis Drieghe.

**Project administration:** Lewis T. Jayes, Gemma Fitzsimmons, Mark J. Weal, Johanna K. Kaakinen, Denis Drieghe.

**Resources:** Lewis T. Jayes, Gemma Fitzsimmons, Mark J. Weal, Johanna K. Kaakinen, Denis Drieghe.

**Software:** Lewis T. Jayes, Gemma Fitzsimmons, Mark J. Weal, Johanna K. Kaakinen, Denis Drieghe.

**Supervision:** Lewis T. Jayes, Gemma Fitzsimmons, Mark J. Weal, Johanna K. Kaakinen, Denis Drieghe.

**Validation:** Lewis T. Jayes, Gemma Fitzsimmons, Mark J. Weal, Johanna K. Kaakinen, Denis Drieghe.

**Visualization:** Lewis T. Jayes, Gemma Fitzsimmons, Mark J. Weal, Johanna K. Kaakinen, Denis Drieghe.

**Writing – original draft:** Lewis T. Jayes, Gemma Fitzsimmons, Mark J. Weal, Johanna K. Kaakinen, Denis Drieghe.

**Writing – review & editing:** Lewis T. Jayes, Gemma Fitzsimmons, Mark J. Weal, Johanna K. Kaakinen, Denis Drieghe.

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
