## [Decision Letter · Decision Letter 0]

23 Sep 2021

PONE-D-21-22263The Impact of Hyperlinks, Skim Reading and Perceived Importance when Reading on the WebPLOS ONE

Dear Dr. Fitzsimmons,

Thank you for submitting your manuscript to PLOS ONE. After careful consideration, we feel that it has merit but does not fully meet PLOS ONE’s publication criteria as it currently stands. Therefore, we invite you to submit a revised version of the manuscript that addresses the points raised during the review process.

As you will see in the comments below, both reviewers believe that your manuscript will make a significant contribution to the empirical literature on reading. However, they raised a number of important concerns, particularly with regard to the statistical analyses (e.g., no inclusion of a power analysis) and clarity (e.g., parts of the introduction require some additional elaboration).

Could you please address these concerns in a revised version of your manuscript by Nov 06 2021 11:59PM. If you will need more time than this to complete your revisions, please reply to this message or contact the journal office at plosone@plos.org. Please include the following items when submitting your revised manuscript:A rebuttal letter that responds to each point raised by the academic editor and reviewer(s). You should upload this letter as a separate file labeled 'Response to Reviewers'.A marked-up copy of your manuscript that highlights changes made to the original version. You should upload this as a separate file labeled 'Revised Manuscript with Track Changes'.An unmarked version of your revised paper without tracked changes. You should upload this as a separate file labeled 'Manuscript'.

We look forward to receiving your revised manuscript.

Kind regards,

Veronica Whitford

Academic Editor

PLOS ONE

“GF was funded by an EPSRC grant for the Doctoral Training Centre in Web Science: EP/G036926/1. This work formed a part of a PhD completed in the Web Science DTC.”

Please include this amended Role of Funder statement in your cover letter; we will change the online submission form on your behalf

Reviewers' comments:

Reviewer's Responses to Questions

**Comments to the Author**

1. Is the manuscript technically sound, and do the data support the conclusions?

Reviewer #1: Yes

Reviewer #2: Yes

2. Has the statistical analysis been performed appropriately and rigorously? 

Reviewer #1: Yes

Reviewer #2: Yes

3. Have the authors made all data underlying the findings in their manuscript fully available?

Reviewer #1: Yes

Reviewer #2: No

4. Is the manuscript presented in an intelligible fashion and written in standard English?

Reviewer #1: Yes

Reviewer #2: Yes

5. Review Comments to the Author

Reviewer #1: Dear Authors,

I really appreciate the opportunity to read and revise your manuscript. I find the topic very interesting, timely, and relevant. Your study offers the opportunity to increase our knowledge in the field of digital reading. I especially appreciate that your work sheds more light on how the presence of hyperlinks affects reading behavior. Your study is well designed, and I also value positively the use of LMM analyses. Moreover, I enjoyed reading your report. However, I think that there are some issues that should be addressed. I hope that my comments help you improve the manuscript and enhance the relevance of your work.

ABSTRACT:

- The use of the eye-tracking technique to measure reading times should be reported in the abstract, not only for the sake of clarity, but also to highlight one of the strengths of this work.

- In my opinion, authors should specify that participants for Study 1 and 2 were not the same. As currently worded, the reader could interpret that it was the same sample.

INTRODUCTION:

- I wonder if the authors could use a more recent reference to support the idea that readers tend to skim when reading on the web, as the current one (Liu, 2005) appears to be somewhat old given the rapid evolution in the widespread use of the Internet.

- I suggest to modify the heading ‘Reading on the Web and Importance’ -> ‘Reading on the Web and Text Importance’

- A brief introduction on the use of the eye-tracking technique as a measure in (digital) reading research could be included at the beginning of the section ‘Reading on the web and importance’. What are the more relevant measures, and what they are assumed to indicate (e.g., attention allocation, strategic reading…). This would be informative for those readers that are not familiar with this methodology.

- p. 3, line 62: “of reading” -> on reading.

- p. 3, line 62: I think that what a “static environment” is should be clarified at his point.

- I miss further elaboration on the possible consequences of the interaction between hyperlinks, perceived importance, and skim reading on the quality of readers’ information processing when reading on the web, as this is the most important practical consequence of these reading practices.

STUDY 1:

Method:

- I miss information on which degrees participants are enrolled in.

- In a footnote, the authors explain that the imbalance between sub-samples’ size is due to the analyses performed. However, it still remains not clear to me. Why the analyses resulted in different group sizes?

- Authors say that the sample sizes are typical of eye-tracking reading research. However, this is not the case for Study 1. The sample size in this study appears somewhat small to me. In my opinion, this should be mentioned as a limitation.

- I think that the experimental setting (i.e., the use of 2 screens presenting the same text differently formatted) could appear somewhat strange to the reader. Although the authors explain in the ‘Procedure’ section how the participants interact with both devices, in my opinion this should be briefly clarified in the ‘Apparatus’ section, namely, that the participants read the sentence-by-sentence text on the laptop after they read the whole text on the desktop computer).

- I miss information on the word length of the texts used.

- When the authors refer to “target words” in ‘Stimuli and Design’ section, do they refer to linked words? As far as I understood, they are not “targets” in this study, so I think that the authors should refer to them, in this case, as hyperlinks instead of target words.

- Why four sentences were inserted into each article?

- If I am not wrong, the hyperlinks consisted of only one word in all cases (i.e., the target words in the previous Fitzsimmons et al.’s study). Whether this is the case or not, the number of words the hyperlinks consisted of should be explicitly mentioned.

- Specify that the number of links were measured per sentence, and the position on the screen was also for each sentence.

- If I understood well, the presence of links was counterbalanced across sentences and participants in the linked passages group. This allows to examine the effect of links regardless of the sentence content, as all sentences are linked taking the four versions altogether. For the sake of clarity, I think that this could be explicitly mentioned.

- Authors could include an “Analyses” subsection including the first paragraph of the ‘Results’ section.

Results:

- Please revise APA standards. Statistical indexes (SD, p… must be italicized)

- P. 11: Why the difference between Passage types may be due to lower rates of the unlinked sentences in the linked texts? Why the authors did not perform between-groups comparisons to examine this possibility?

- I think that Figure 2 it is not fully self-explanatory. I understand that different boxes correspond to different positions on page, but this is not well clarified in the figure. The same with Figure 6 and length of sentence.

- P. 13: I think there is something wrong when reporting descriptive data for the ratings of linked vs unlinked sentences. Mean and SD of the ratings for the whole linked texts in the previous section (‘Effect of Hyperlinks’) are the same than those for only the unlinked sentences in such texts. This is not possible, given that the ratings for the linked sentences are higher.

- The scale for the independent continuous variables in the figures show negative values because the authors centered them. However, I suggest to change these scales in the figures by including not centered values. I think this would make it easy to visually interpret the results to the reader.

- Please, revise the variable names in the figures, e.g., Position_on_Page -> Position on page.

Discussion:

- As I mentioned earlier, I do not fully understand which result the authors based on to conclude that the unlinked sentences were rated lower in the presence of other linked sentences than the same sentences in unlinked texts. The authors did not compare differences in the rating for unlinked sentences between passage-type groups, however they state in the discussion that: “When a sentence does not contain links but is in a hypertext environment with other sentences that do contain links, it is rated as being of lower importance”.

- P 17. Authors conclude that “long sentences with many links were rated higher, no matter the position of the sentence on the page. This suggests that links do make a sentence more important, especially for long sentences”. This conclusion is based on the two-way interaction between sentence length and position. As far as I understood, the fact that long sentences were rated lower at the end of the page, but higher when they contain links, supports this conclusion. I infer that authors meant this when saying ‘when a sentence near the bottom of the page would be rated quite low, if it contains a link it gets a boost of importance”. But I think that the interaction should be explicitly mentioned here.

- P. 18. Authors also conclude that “readers recognise the “inverted pyramid” structure of webpages, when providing offline ratings of importance”. I think that the authors did not controlled whether the most relevant content was actually located at the top of the texts used in the study, thus, in my opinion, it is not possible to conclude that the participants “recognised” the inverted pyramid structure.

EXPERIMENT 2:

Intro:

- P.19. Authors wrote: “By using links, the reader can efficiently identify important information in the text and move through the text faster…”. However, it depends on whether the links are located in the sentences including important information, which is not always the case on the web. In my opinion, the fact that locating links in non-relevant sentences can mislead the reader and, therefore, hinder comprehension should be further discussed, especially in the ‘Discussion’ section.

- P.21: I think that the direction of the association between the wrap-up effect and the other variables (information density, processing capacity, literacy development) should be clarified. Are they negatively or positively associated?

Method:

- Please, provide more information about the comprehension questions. Did they address text-based or inferential comprehension? Also, providing some examples of the questions would be illustrative.

Results:

- In my opinion, the experimental measures should be previously reported in the Method section. The same with the ‘Analyses’.

- When reporting the measures, please specify what the authors considered as “first pass reading”.

- Why the authors did not constrain their analyses to reading time per character, as this is the standard measure in eye-tracking reading research? Otherwise, reading times and sentence length are confounded.

- P. 31. Please include spaces between statistical letters, symbols, and values when reporting results from ANOVA.

- P. 31. The authors say that “…which suggests that the participants were to a degree engaged in a successful adaptive strategy because they had improved accuracy for comprehension questions relating to the most important information.” I would avoid the use of “successful” as it depends on whether the sentences identified as important are actually important based on the text content. Results from Study 1 show that sentence importance is rated based on features other than semantic content, thus an adaptive strategy would depend on, for example, whether links are located in semantically relevant sentences. Indeed, I would say that a reading strategy based on the presence of links seems to be less effective than relying on semantic content. As I mentioned before, links are often located in less relevant sentences on the web. Indeed, links often serve to provide additional, complementary information, hence secondary for the text content.

Discussion:

- I think that the authors did not sufficiently addressed the fact that, in this study, online measures showed that readers were not sensitive when skim reading to the signals they use as indicators of relevance when reading for comprehension. In this regard, this result is contrary to those in previous research discussed in the introduction of the manuscript (p. 4: “These findings suggested readers were prioritising the processing of visually salient words while skim reading webpage”).

- In relation to the above, authors conclude on p. 33 that: “Given sentences with more links were rated as more important, and skim reading leads to increased focus on links [1], it seems readers use links as signals through the text to anchor attention, leading to increased comprehension of those sentences.” This conclusion is based on previous results that are in conflict with findings in the current study, as I noted above.

- I think that the fact that readers had too broad reading goals (i.e., reading for comprehension or reading for skimming) could influence their effort in detecting important parts of the texts when skim reading. It is possible that a more specific reading goal would have encouraged them to identify important parts also when skimming and this could have been reflected in their reading behavior. I would mention this as a limitation of the study.

GENERAL DISCUSSION

- In p. 35 it is concluded that “it could be suggested that the multi-faceted signal functions of hyperlinks cause them to be considered as a more important typographical cue than others”. In my opinion, a conclusion like that needs to be based on an experimental comparison between the influence of hyperlinks and, for example, the influence of mere highlighting. I suggest to mention this in the manucript.

- My main concern about the General Discussion is that I think that the authors should elaborate on the fact that readers rely on physical cues regardless of semantic content which could impact negatively reading comprehension when reading on the web. As I noted above, the presence of links is not a guarantee of content relevance, but sometimes they are located in sentences that are secondary for the text main content to provide additional info. Other times, links are even located in random words to link to commercial websites. Thus, the findings of this study have important educational implications that must be considered when training students to gather information from the web.

- In relation to the above, I think that it could be suggested that further research is needed to examine to what extent readers base their use of reading strategies on cues other than semantic relevance, which must ultimately be the cues on which a good (digital) reader must base their decisions.

- I also miss further discussion on why readers consider longer sentences as more important only at the top of the page and, conversely, shorter sentences are considered more important at the bottom. If I may suggest, a possible explanation is that readers’ attention or allocation of cognitive effort decreases as they move down the webs and they then focus on shorter sentences. As they focus more on these sentences and, consequently, they comprehend them better, this could be an explanation for an increased perceived importance.

Reviewer #2: Dear Prof. Whitford,

The manuscript entitled " The Impact of Hyperlinks, Skim Reading and Perceived Importance when Reading on the Web." is an empirical paper investigating the effect of highlighting and text position on the intelligibility of a text. The authors implemented two studies, one rating study investigating subjective importance ratings of sentences and a second study, based on the ratings from study one, that investigates specific online reading behaviors with eye-tracking methodology. The relatively complex analysis required careful implementation and design. The study resulted in several highly interesting findings, such as the influence of highlighting, text position, and sentence length. Besides, it showed that the latter effects were significantly reduced when participants implemented fast skimming compared to regular reading. The study is a highly valuable piece of evidence, but several issues prevent me from recommending the manuscript for publication. Please find my specific comments below. Besides, I would recommend making data and code related to the paper openly available.

Best wishes,

Benjamin Gagl

#Comments

##Statistical power. Please provide argumentation for the sample size based on statistical power. If no a-priory power estimation was implemented, please give a post-hoc assessment for a range of potentially interesting effect sizes. Such an analysis would be highly informative for future replication efforts. Please note, recent developments now provide convenient tools for power estimations based on LMM models (e.g., see Green, P., & MacLeod, C. J. (2016). SIMR: an R package for power analysis of generalized linear mixed models by simulation. Methods in Ecology and Evolution, 7(4), 493-498. or Kumle, L., Vo, M. L. H., & Draschkow, D. (2021). Estimating power in (generalized) linear mixed models: An open introduction and tutorial in R. Behavior Research Methods, 1-16.).

##More details on the stimuli. Considerable detail is missing in the description of the stimuli. Starting with the N of trials/sentences/paragraphs, or potential informativeness measures the sentences (e.g., number of content words). Also, characteristics of hyperlinked words (lengths, status, frequency etc.). Besides, it is currently not clear to me how one can end up with a sentence length or position on a page that is negative (e.g., see Figure 2/3). Would you mind providing more information here.

##Presentation of results in general. From a presentation standpoint, it makes sense to start with higher-order interactions first. This is as only resolving them first would make the statistical results fully accessible. Similar to your procedure for a parsimonious random effect structure. Also, one could, at times, remove figures that depict the lower-order interactions to simplify the presentation (e.g., Fig. 1/7)

##p. 12 270-272: Please revisit this sentence. It is currently unclear which terms were excluded based on this sentence. Would you please elaborate on the procedure in more detail.

##Elaborate on the differences between skim and normal reading. E.g., the role of para/extra-foveal processing of text (e.g., as investigated in Gagl, 2016).

##Details of the eye-movement measurement procedure. E.g., calibration procedure, fixation detection algorithm, etc. The current description is missing detail here. For guidance, see the parameter list in the BEP-20 extension proposal of the BIDS data standard (https://docs.google.com/document/d/1eggzTCzSHG3AEKhtnEDbcdk-2avXN6I94X8aUPEBVsw/edit#heading=h.1yd8ejg9cdbt). Also, comments are welcome!

#Minors

##Please harmonize the description of random effects, e.g., cp. Description in Table 1 and 3.

##Figure 8/9. Why is the figure style different from, e.g., Fig.10 (i.e., one figure with different line coloring). Using the same style would help to see the differences in the interactions better.

6. PLOS authors have the option to publish the peer review history of their article (what does this mean?). If published, this will include your full peer review and any attached files.

Reviewer #1: No

Reviewer #2: **Yes: **Benjamin Gagl

---

## [Author Response · Author response to Decision Letter 0]

9 Nov 2021

Thank you for your consideration of our submission. In line with the constructive feedback from the reviewers we have revised the manuscript and would like to resubmit it for further consideration, as per your invitation. 

Please find below our responses to the reviewers comment. We hope that you will find our replies and revisions to satisfactorily address yours and the reviewers’ concerns. 

Reviewer #1: Dear Authors,

I really appreciate the opportunity to read and revise your manuscript. I find the topic very interesting, timely, and relevant. Your study offers the opportunity to increase our knowledge in the field of digital reading. I especially appreciate that your work sheds more light on how the presence of hyperlinks affects reading behavior. Your study is well designed, and I also value positively the use of LMM analyses. Moreover, I enjoyed reading your report. However, I think that there are some issues that should be addressed. I hope that my comments help you improve the manuscript and enhance the relevance of your work.

We thank the reviewer for their useful comments in improving the manuscript and provide our replies and edits for each individual comment below.

ABSTRACT:

- The use of the eye-tracking technique to measure reading times should be reported in the abstract, not only for the sake of clarity, but also to highlight one of the strengths of this work.

We now make the use of eye tracking clearer in the abstract (page 2, lines 25 and 33-34)

- In my opinion, authors should specify that participants for Study 1 and 2 were not the same. As currently worded, the reader could interpret that it was the same sample.

We now mention this detail in the abstract (line 24)

INTRODUCTION:

- I wonder if the authors could use a more recent reference to support the idea that readers tend to skim when reading on the web, as the current one (Liu, 2005) appears to be somewhat old given the rapid evolution in the widespread use of the Internet.

More recent references are now provided to support Liu’s statement, detailing the cost to processing that skimming and scanning causes (Lenhard et al., 2017; Singer & Alexander, 2017; Salmeron et al., 2016) – see lines 55-59, page 3

- I suggest to modify the heading ‘Reading on the Web and Importance’ -> ‘Reading on the Web and Text Importance’

This subheading has been amended as the reviewer suggests, to reduce ambiguity (line 64, page 3)

- A brief introduction on the use of the eye-tracking technique as a measure in (digital) reading research could be included at the beginning of the section ‘Reading on the web and importance’. What are the more relevant measures, and what they are assumed to indicate (e.g., attention allocation, strategic reading…). This would be informative for those readers that are not familiar with this methodology.

The opening paragraph of this section now details the use of eye movement measures to investigate reading behaviour, how this has been extended to investigate digital reading, and how eye movement measures are typically utilised in this field (lines 65-85, page 4-5)

- p. 3, line 62: “of reading” -> on reading

Corrected 

- p. 3, line 62: I think that what a “static environment” is should be clarified at his point.

Definition of static environment is now provided (line 88, page 5)

- I miss further elaboration on the possible consequences of the interaction between hyperlinks, perceived importance, and skim reading on the quality of readers’ information processing when reading on the web, as this is the most important practical consequence of these reading practices.

We now mention this in the introduction to the studies (lines 167-169), which also links to our applications section within the general discussion, where we also elaborate on this point further (lines 1083-1090, pages 38)

STUDY 1:

Method:

- I miss information on which degrees participants are enrolled in.

This detail is now provided

- In a footnote, the authors explain that the imbalance between sub-samples’ size is due to the analyses performed. However, it still remains not clear to me. Why the analyses resulted in different group sizes?

We analysed the linked dataset to investigate the effects of number of links, position on page and sentence length. This was not necessary for the unlinked dataset, which only served as a comparison dataset for the linked data for one set of analyses. As such, more participants were required to provide the requisite power to analyse these variables. This is now clarified in the footnote (lines 1358, page 49).

- Authors say that the sample sizes are typical of eye-tracking reading research. However, this is not the case for Study 1. The sample size in this study appears somewhat small to me. In my opinion, this should be mentioned as a limitation.

Sample sizes like that used in this study are typical of the field. Please see our additional power analyses, carried out on the recommendation of the second reviewer, indicating our sample size was adequate for our analyses. 

- I think that the experimental setting (i.e., the use of 2 screens presenting the same text differently formatted) could appear somewhat strange to the reader. Although the authors explain in the ‘Procedure’ section how the participants interact with both devices, in my opinion this should be briefly clarified in the ‘Apparatus’ section, namely, that the participants read the sentence-by-sentence text on the laptop after they read the whole text on the desktop computer).

This is now clarified in the procedure section (lines 290-291, page 11)

- I miss information on the word length of the texts used.

This is clarified on a word level (line 254-257, page 10) and sentence level (line 230, page 9)

- When the authors refer to “target words” in ‘Stimuli and Design’ section, do they refer to linked words? As far as I understood, they are not “targets” in this study, so I think that the authors should refer to them, in this case, as hyperlinks instead of target words.

We have removed mention target words with regard to the current study. We referred to target words as this was how they were regarded when the stimuli were developed in the original Fitzsimmons et al., paper. 

- Why four sentences were inserted into each article?

This detail explains how the Wikipedia pages were adapted. We endeavoured to retain as much of their original form as possible. In order to operationalise previous manipulations of frequency and length (see Fitzsimmons et al., 2019), sentences that used manipulated target words were inserted into the stimuli. This detail is provided to ensure we are as explicit as possible about any changes we made to the original Wikipedia pages. This detail is expanded upon in the stimuli and design section.

- If I am not wrong, the hyperlinks consisted of only one word in all cases (i.e., the target words in the previous Fitzsimmons et al.’s study). Whether this is the case or not, the number of words the hyperlinks consisted of should be explicitly mentioned.

This detail is now provided explicitly (line 244, page 10)

- Specify that the number of links were measured per sentence, and the position on the screen was also for each sentence.

We now specify this detail when discussing the variables for Experiment 1 (line 244, page 10)

- If I understood well, the presence of links was counterbalanced across sentences and participants in the linked passages group. This allows to examine the effect of links regardless of the sentence content, as all sentences are linked taking the four versions altogether. For the sake of clarity, I think that this could be explicitly mentioned.

We now explicitly state this (line 262, page 10)

- Authors could include an “Analyses” subsection including the first paragraph of the ‘Results’ section.

The results section now has an ‘ Analyses’ subheading at the beginning (line 297, page 11)

Results:

- Please revise APA standards. Statistical indexes (SD, p… must be italicized)

This has been reviewed and adjusted across the results sections. 

- P. 11: Why the difference between Passage types may be due to lower rates of the unlinked sentences in the linked texts? Why the authors did not perform between-groups comparisons to examine this possibility?

The main effect within this model of passage type is a between subjects comparison of participants who completed the ratings task on the linked or unlinked stimuli, ensuring we did examine this possibility and report it. We have clarified this detail when discussing this analysis to ensure this is clearer (line 323, page 12). 

- I think that Figure 2 it is not fully self-explanatory. I understand that different boxes correspond to different positions on page, but this is not well clarified in the figure. The same with Figure 6 and length of sentence.

We have clarified this in the figure caption to remove ambiguity (line 351, page 13, lines 505, page 17) 

- P. 13: I think there is something wrong when reporting descriptive data for the ratings of linked vs unlinked sentences. Mean and SD of the ratings for the whole linked texts in the previous section (‘Effect of Hyperlinks’) are the same than those for only the unlinked sentences in such texts. This is not possible, given that the ratings for the linked sentences are higher.

These two analyses are of different datasets, the unlinked task dataset (as detailed on line 323), were excluded from the second analysis. In the first analysis, unlinked is referring to ratings from participants who completed the task on completely unlinked text. In the second analysis, without links refers to individual sentences that did not feature links, but only for those who completed the task on linked passages. As such, the two numbers are not comparable. 

- The scale for the independent continuous variables in the figures show negative values because the authors centered them. However, I suggest to change these scales in the figures by including not centered values. I think this would make it easy to visually interpret the results to the reader.

We now depict all figures using the raw data, rather than the centred variables

- Please, revise the variable names in the figures, e.g., Position_on_Page -> Position on page.

This revision has been made to the variable names across all figures

Discussion:

- As I mentioned earlier, I do not fully understand which result the authors based on to conclude that the unlinked sentences were rated lower in the presence of other linked sentences than the same sentences in unlinked texts. The authors did not compare differences in the rating for unlinked sentences between passage-type groups, however they state in the discussion that: “When a sentence does not contain links but is in a hypertext environment with other sentences that do contain links, it is rated as being of lower importance”.

We did compare differences in the ratings for unlinked sentences between passage group types, as this is what the main effect of passage type in this analysis was. We have clarified this in the results section (as detailed above) and have clarified further in the discussion (line 533, page 18). 

- P 17. Authors conclude that “long sentences with many links were rated higher, no matter the position of the sentence on the page. This suggests that links do make a sentence more important, especially for long sentences”. This conclusion is based on the two-way interaction between sentence length and position. As far as I understood, the fact that long sentences were rated lower at the end of the page, but higher when they contain links, supports this conclusion. I infer that authors meant this when saying ‘when a sentence near the bottom of the page would be rated quite low, if it contains a link it gets a boost of importance”. But I think that the interaction should be explicitly mentioned here.

We have clarified the interactions that we used to support the findings we discuss in this section of the discussion (line 594 and 600, page 19). 

- P. 18. Authors also conclude that “readers recognise the “inverted pyramid” structure of webpages, when providing offline ratings of importance”. I think that the authors did not controlled whether the most relevant content was actually located at the top of the texts used in the study, thus, in my opinion, it is not possible to conclude that the participants “recognised” the inverted pyramid structure.

We have reworded this statement to reduce the claim that it was recognised, and to focus more on the face that offline ratings seem to mirror this type of structure. 

EXPERIMENT 2:

Intro:

- P.19. Authors wrote: “By using links, the reader can efficiently identify important information in the text and move through the text faster…”. However, it depends on whether the links are located in the sentences including important information, which is not always the case on the web. In my opinion, the fact that locating links in non-relevant sentences can mislead the reader and, therefore, hinder comprehension should be further discussed, especially in the ‘Discussion’ section.

This issue is now discussed in the discussion (lines 1050, page 37). We have also altered the above sentence to caveat that this detail is more relevant when hyperlinks are used appropriately, based on previous research in Fitzsimmons, Weal & Drieghe, 2019. 

- P.21: I think that the direction of the association between the wrap-up effect and the other variables (information density, processing capacity, literacy development) should be clarified. Are they negatively or positively associated?

We have changed “increase as a function” to “positively associated” to provide further clarity (line 711, page 23)

Method:

- Please, provide more information about the comprehension questions. Did they address text-based or inferential comprehension? Also, providing some examples of the questions would be illustrative.

The questions were text-based questions, designed to ensure readers were engaged, comprehending the text and reading naturally. We now clarify this and provide an example (line 790, page 26) 

Results:

- In my opinion, the experimental measures should be previously reported in the Method section. The same with the ‘Analyses’.

This could be considered atypical of eye movement studies, please see our previously published eye movement studies in PLOS One (Fitzsimmons, Weal & Drieghe, 2019; Fitzsimmons, Jayes, Weal & Drieghe, 2020) for examples of this method and results section. Furthermore, this does not functionally alter the structure of the manuscript, as the measures and analyses come after the method and before the results are presented. 

- When reporting the measures, please specify what the authors considered as “first pass reading”.

A definition of first pass is now provided (line 814, page 27)

- Why the authors did not constrain their analyses to reading time per character, as this is the standard measure in eye-tracking reading research? Otherwise, reading times and sentence length are confounded.

The measures we use in our study are extremely typical of eye movement and reading research (see Fitzsimmons, Weal & Drieghe, 2019; Fitzsimmons, Jayes, Weal & Drieghe, 2020 and Rayner, 1998; 2009 for review). Furthermore, we provide per character analyses to show that this confound does not affect the pattern of results we found. By not just using per character analyses, we were able to analyse the time course of how readers read these sentences. For example, it means we can investigate the differences between first pass and total reading time. Total reading time allowed us to investigate if refixations beyond the first pass are impacted upon by importance and task. As such, we include both the first pass and total reading times, with the per character measure provided to ensure the confound of sentence length is not affecting our results, as the pattern of results is replicated. 

- P. 31. Please include spaces between statistical letters, symbols, and values when reporting results from ANOVA.

Spaces are now included in the reporting of this result (lines 974-979, pages 34)

- P. 31. The authors say that “…which suggests that the participants were to a degree engaged in a successful adaptive strategy because they had improved accuracy for comprehension questions relating to the most important information.” I would avoid the use of “successful” as it depends on whether the sentences identified as important are actually important based on the text content. Results from Study 1 show that sentence importance is rated based on features other than semantic content, thus an adaptive strategy would depend on, for example, whether links are located in semantically relevant sentences. Indeed, I would say that a reading strategy based on the presence of links seems to be less effective than relying on semantic content. As I mentioned before, links are often located in less relevant sentences on the web. Indeed, links often serve to provide additional, complementary information, hence secondary for the text content.

We agree and have removed the use of the word ‘successful’ (line 977, page 34)

Discussion:

- I think that the authors did not sufficiently addressed the fact that, in this study, online measures showed that readers were not sensitive when skim reading to the signals they use as indicators of relevance when reading for comprehension. In this regard, this result is contrary to those in previous research discussed in the introduction of the manuscript (p. 4: “These findings suggested readers were prioritising the processing of visually salient words while skim reading webpage”).

We now discuss this point (line 1013, page 346. We also now direct readers to our discussion of this aspect in the General Discussion.

- In relation to the above, authors conclude on p. 33 that: “Given sentences with more links were rated as more important, and skim reading leads to increased focus on links [1], it seems readers use links as signals through the text to anchor attention, leading to increased comprehension of those sentences.” This conclusion is based on previous results that are in conflict with findings in the current study, as I noted above.

This result is a discussion of comprehension results, not eye movement measures. This result concerns the fact that despite a lack of use of importance to direct attention during skim reading, comprehension still seems to improve marginally for more important sentences. Furthermore, previous research cited here was conducted at a word level, whereas our analyses were conducted at a sentence-level, meaning we cannot say they are completely contrary, but do work together to build up our understanding of the current phenomena. We have reworded this section to remove this ambiguity (line 1020, pages 36). 

- I think that the fact that readers had too broad reading goals (i.e., reading for comprehension or reading for skimming) could influence their effort in detecting important parts of the texts when skim reading. It is possible that a more specific reading goal would have encouraged them to identify important parts also when skimming and this could have been reflected in their reading behavior. I would mention this as a limitation of the study.

This limitation and need for further research in task effects is now discussed (lines 1101, pages 39), specifically in the context of past task effect research and its implications for SARA. 

GENERAL DISCUSSION

- In p. 35 it is concluded that “it could be suggested that the multi-faceted signal functions of hyperlinks cause them to be considered as a more important typographical cue than others”. In my opinion, a conclusion like that needs to be based on an experimental comparison between the influence of hyperlinks and, for example, the influence of mere highlighting. I suggest to mention this in the manucript.

We now mention this point and reference the experimental work that exists exploring this particular comparison for support (lines 1086, page 37)

- My main concern about the General Discussion is that I think that the authors should elaborate on the fact that readers rely on physical cues regardless of semantic content which could impact negatively reading comprehension when reading on the web. As I noted above, the presence of links is not a guarantee of content relevance, but sometimes they are located in sentences that are secondary for the text main content to provide additional info. Other times, links are even located in random words to link to commercial websites. Thus, the findings of this study have important educational implications that must be considered when training students to gather information from the web.

We now mention this issue more in the discussion (line 1101, page 37) and have introduced this issue into the current discussion of the interaction between typographical and semantic cues in affected perceived importance of text.

- In relation to the above, I think that it could be suggested that further research is needed to examine to what extent readers base their use of reading strategies on cues other than semantic relevance, which must ultimately be the cues on which a good (digital) reader must base their decisions.

We now link this element into our discussion of the above comment (lines 1101, page 38). 

- I also miss further discussion on why readers consider longer sentences as more important only at the top of the page and, conversely, shorter sentences are considered more important at the bottom. If I may suggest, a possible explanation is that readers’ attention or allocation of cognitive effort decreases as they move down the webs and they then focus on shorter sentences. As they focus more on these sentences and, consequently, they comprehend them better, this could be an explanation for an increased perceived importance.

We refrained from drawing too much from this finding, as it is qualified by a higher order interaction with number of links (which we now elaborate on in the text). We elaborate more on this finding in the discussion of experiment 1, discussing this finding in relation to information bias (lines 626, page 20).

Reviewer #2: Dear Prof. Whitford,

The manuscript entitled " The Impact of Hyperlinks, Skim Reading and Perceived Importance when Reading on the Web." is an empirical paper investigating the effect of highlighting and text position on the intelligibility of a text. The authors implemented two studies, one rating study investigating subjective importance ratings of sentences and a second study, based on the ratings from study one, that investigates specific online reading behaviors with eye-tracking methodology. The relatively complex analysis required careful implementation and design. The study resulted in several highly interesting findings, such as the influence of highlighting, text position, and sentence length. Besides, it showed that the latter effects were significantly reduced when participants implemented fast skimming compared to regular reading. The study is a highly valuable piece of evidence, but several issues prevent me from recommending the manuscript for publication. Please find my specific comments below. Besides, I would recommend making data and code related to the paper openly available.

Best wishes,

Benjamin Gagl

We thank the reviewer for their useful comments in improving the manuscript and provide our replies and edits for each individual comment below. Please note we have made all data and code freely available and open access on the UK data service, please find the reference for this below:

Fitzsimmons, G. (2021). The Impact of Hyperlinks, Skim Reading and Perceived Importance when Reading on the Web, 2009-2019. [data collection]. UK Data Service. SN: 855044, http://doi.org/10.5255/UKDA-SN-855044

#Comments

##Statistical power. Please provide argumentation for the sample size based on statistical power. If no a-priory power estimation was implemented, please give a post-hoc assessment for a range of potentially interesting effect sizes. Such an analysis would be highly informative for future replication efforts. Please note, recent developments now provide convenient tools for power estimations based on LMM models (e.g., see Green, P., & MacLeod, C. J. (2016). SIMR: an R package for power analysis of generalized linear mixed models by simulation. Methods in Ecology and Evolution, 7(4), 493-498. or Kumle, L., Vo, M. L. H., & Draschkow, D. (2021). Estimating power in (generalized) linear mixed models: An open introduction and tutorial in R. Behavior Research Methods, 1-16.).

We now explicitly discuss the sample sizes used for this study, with discussion of previous studies, as well as our post-hoc power analyses using SIMR, which consistently returned power above 80% for the significant effects discussed within the manuscript. 

##More details on the stimuli. Considerable detail is missing in the description of the stimuli. Starting with the N of trials/sentences/paragraphs, or potential informativeness measures the sentences (e.g., number of content words). Also, characteristics of hyperlinked words (lengths, status, frequency etc.). Besides, it is currently not clear to me how one can end up with a sentence length or position on a page that is negative (e.g., see Figure 2/3). Would you mind providing more information here.

We now clarify the details of the stimuli across the stimuli and design section of Experiment 1, including length and frequency detail (lines 254). As stated in the manuscript, the stimuli are taken from Fitzsimmons et al. (2019), where more information can be found relating to the stimuli. We now highlight this in the stimuli section (lines 281) and encourage readers to find the full details of the individual word analyses there. 

##Presentation of results in general. From a presentation standpoint, it makes sense to start with higher-order interactions first. This is as only resolving them first would make the statistical results fully accessible. Similar to your procedure for a parsimonious random effect structure. 

We have now adapted the results of both experiments 1 and 2 to present the higher order interactions, before detailing the main effects. 

Also, one could, at times, remove figures that depict the lower-order interactions to simplify the presentation (e.g., Fig. 1/7)

We have removed figure 1 from the manuscript, as it is possible to see the lower order interaction in figure 2. We have refrained from removing figure 7, as we believe it is not easily possible to understand the interaction just from figure 6, due to the multiple levels and the continuous nature of the variables. 

##p. 12 270-272: Please revisit this sentence. It is currently unclear which terms were excluded based on this sentence. Would you please elaborate on the procedure in more detail.

We have clarified this point

##Elaborate on the differences between skim and normal reading. E.g., the role of para/extra-foveal processing of text (e.g., as investigated in Gagl, 2016).

We now discuss the role of skipping and foveal/parafoveal processing in relation to hyperlinks, basing this discussion on the findings of Gagl (2016) (lines 92, pages 4)

##Details of the eye-movement measurement procedure. E.g., calibration procedure, fixation detection algorithm, etc. The current description is missing detail here. For guidance, see the parameter list in the BEP-20 extension proposal of the BIDS data standard (https://docs.google.com/document/d/1eggzTCzSHG3AEKhtnEDbcdk-2avXN6I94X8aUPEBVsw/edit#heading=h.1yd8ejg9cdbt). Also, comments are welcome!

We provide further detail of the calibration procedure in the method section (line 780) and detail of fixation detection algorithm at the beginning of the results (lines 800-810)

#Minors

##Please harmonize the description of random effects, e.g., cp. Description in Table 1 and 3.

We have reviewed the tables throughout the manuscript and sought to harmonize and standardize the descriptions. 

##Figure 8/9. Why is the figure style different from, e.g., Fig.10 (i.e., one figure with different line coloring). Using the same style would help to see the differences in the interactions better.

We have adapted these figures to follow the same style as the other figures.

---

## [Decision Letter · Decision Letter 1]

12 Jan 2022

PONE-D-21-22263R1 The Impact of Hyperlinks, Skim Reading and Perceived Importance when Reading on the WebPLOS ONE

Dear Dr. Fitzsimmons,

Thank you once for submitting your manuscript to PLOS ONE. As you will see below, both Reviewers 1 and 2 agree that your manuscript was considerably strengthened by the last round of revisions and that you have satisfactorily addressed their major concerns. However, they have noted some relatively minor issues that would need to be addressed before the manuscript is accepted for publication. Could you please submit the revised version by March 11, 2022? If you will need more time than this to complete your revisions, please reply to this message or contact the journal office at plosone@plos.org. Please include the following items when submitting your revised manuscript:A rebuttal letter that responds to each point raised by the academic editor and reviewer(s). You should upload this letter as a separate file labeled 'Response to Reviewers'.A marked-up copy of your manuscript that highlights changes made to the original version. You should upload this as a separate file labeled 'Revised Manuscript with Track Changes'.An unmarked version of your revised paper without tracked changes. You should upload this as a separate file labeled 'Manuscript'.If applicable, we recommend that you deposit your laboratory protocols in protocols.io to enhance the reproducibility of your results. Protocols.io assigns your protocol its own identifier (DOI) so that it can be cited independently in the future. For instructions see: https://journals.plos.org/plosone/s/submission-guidelines#loc-laboratory-protocols. Additionally, PLOS ONE offers an option for publishing peer-reviewed Lab Protocol articles, which describe protocols hosted on protocols.io. Read more information on sharing protocols at https://plos.org/protocols?utm_medium=editorial-email&utm_source=authorletters&utm_campaign=protocols.

We look forward to receiving your revised manuscript.

Kind regards,

Veronica Whitford, Ph.D.

Academic Editor

PLOS ONE

Reviewers' comments:

Reviewer's Responses to Questions

**Comments to the Author**

1. If the authors have adequately addressed your comments raised in a previous round of review and you feel that this manuscript is now acceptable for publication, you may indicate that here to bypass the “Comments to the Author” section, enter your conflict of interest statement in the “Confidential to Editor” section, and submit your "Accept" recommendation.

Reviewer #1: (No Response)

Reviewer #2: All comments have been addressed

2. Is the manuscript technically sound, and do the data support the conclusions?

Reviewer #1: Yes

Reviewer #2: Yes

3. Has the statistical analysis been performed appropriately and rigorously? 

Reviewer #1: Yes

Reviewer #2: Yes

4. Have the authors made all data underlying the findings in their manuscript fully available?

Reviewer #1: Yes

Reviewer #2: Yes

5. Is the manuscript presented in an intelligible fashion and written in standard English?

Reviewer #1: Yes

Reviewer #2: Yes

6. Review Comments to the Author

Reviewer #1: I think that you satisfactorily addressed my comments and your decisions on not following some of them were satisfactorily justified. I only have some additional minor comments. Once again, thank you for giving me the opportunity to help you improve your manuscript:

- I think that total number of words of each text should be reported.

- Authors say that all links were a single word, but I can see in the provided stimuli example, there were links consisting of more than one word.

- Spaces between “F” and the “d.f.” parenthesis must be removed in the statistical results from ANOVAs.

- The variable “Position on page” is still written as “Position_on_page” in Figure 1.

- This is only a suggestion. When discussing on the fact that links are not always relevant on the web, I would also mention those cases in which links bring the reader to some information that is related to the text, but it is not relevant to its main ideas (as it is the case of Wikipedia). If I am not wrong, authors currently mention only those links inserted for commercial purposes.

- I am really sorry, but I couldn’t find the following, maybe due to the fact that page and line numbers provided in your responses don’t match those in the pdf that I downloaded. See my comment and your response: In p. 35 it is concluded that “it could be suggested that the multi-faceted signal functions of hyperlinks cause them to be considered as a more important typographical cue than others”. In my opinion, a conclusion like that needs to be based on an experimental comparison between the influence of hyperlinks and, for example, the influence of mere highlighting. I suggest to mention this in the manuscript.

We now mention this point and reference the experimental work that exists exploring this particular comparison for support (lines 1086, page 37)

Reviewer #2: Dear Prof. Whitford,

The revised version of the manuscript entitled " The Impact of Hyperlinks, Skim Reading and Perceived Importance when Reading on

the Web " considered all of my concerns. The quality of the manuscript increased considerably, but a few minor issues prevent me from recommending the manuscript for publication yet. Please find my specific comments below.

Best wishes,

Benjamin Gagl

#Minor Issues

##Statistical power. The authors now provide a post-power analysis, but an essential parameter for the interpretation of the power analysis is missing, i.e., the effect size. Several effect sizes are presented in the results section, so it would be interesting which effect sizes have been used. Also, if multiple effect sizes have been used, please provide a range for the resulting power values.

##High vs. low frequency comparison (p11; L 245). Would you mind not estimating statistics regarding differences in stimulus characteristics? This procedure has been identified as problematic, and the problem is described in detail here: Sassenhagen, J., & Alday, P. M. (2016). A common misapplication of statistical inference: Nuisance control with null-hypothesis significance tests. Brain and language, 162, 42-45.

Instead, a value range including how much overlap was present would be more informative.

##L12 p. 1: Line sweep missing.

7. PLOS authors have the option to publish the peer review history of their article (what does this mean?). If published, this will include your full peer review and any attached files.

Reviewer #1: No

Reviewer #2: **Yes: **Benjamin Gagl

---

## [Author Response · Author response to Decision Letter 1]

17 Jan 2022

We thank the reviewers for their useful comments in improving the manuscript and provide our replies and edits for each individual comment below.

6. Review Comments to the Author

Reviewer #1: I think that you satisfactorily addressed my comments and your decisions on not following some of them were satisfactorily justified. I only have some additional minor comments. Once again, thank you for giving me the opportunity to help you improve your manuscript:

- I think that total number of words of each text should be reported.

We now report this range on page 9 (line 211-212).

- Authors say that all links were a single word, but I can see in the provided stimuli example, there were links consisting of more than one word.

When discussing one word hyperlinks, we were referring to the hyperlinks inserted into the text, which were tightly controlled (see Fitzsimmons et al, 2019). We also retained all other links in the text, which did differ between 1-3 words. We know note the retention of these links in the text, along with the number of words they consisted of (page 10, line 220).

- Spaces between “F” and the “d.f.” parenthesis must be removed in the statistical results from ANOVAs.

We have removed these spaces (page 34, lines 749-755)

- The variable “Position on page” is still written as “Position_on_page” in Figure 1.

We have changed the variable titles for figure 1 to resolve this.

- This is only a suggestion. When discussing on the fact that links are not always relevant on the web, I would also mention those cases in which links bring the reader to some information that is related to the text, but it is not relevant to its main ideas (as it is the case of Wikipedia). If I am not wrong, authors currently mention only those links inserted for commercial purposes.

We agree and now mention this other aspect of links on Wikipedia (page 39, lines 881-884) 

- I am really sorry, but I couldn’t find the following, maybe due to the fact that page and line numbers provided in your responses don’t match those in the pdf that I downloaded. See my comment and your response: In p. 35 it is concluded that “it could be suggested that the multi-faceted signal functions of hyperlinks cause them to be considered as a more important typographical cue than others”. In my opinion, a conclusion like that needs to be based on an experimental comparison between the influence of hyperlinks and, for example, the influence of mere highlighting. I suggest to mention this in the manuscript.

We now mention this point and reference the experimental work that exists exploring this particular comparison for support (lines 1086, page 37)

This point is made on page 38, lines 854-857, we have edited the paragraph to make it more relevant to the point outlined by the reviewer. 

Reviewer #2: Dear Prof. Whitford,

The revised version of the manuscript entitled " The Impact of Hyperlinks, Skim Reading and Perceived Importance when Reading on

the Web " considered all of my concerns. The quality of the manuscript increased considerably, but a few minor issues prevent me from recommending the manuscript for publication yet. Please find my specific comments below.

Best wishes,

Benjamin Gagl

#Minor Issues

##Statistical power. The authors now provide a post-power analysis, but an essential parameter for the interpretation of the power analysis is missing, i.e., the effect size. Several effect sizes are presented in the results section, so it would be interesting which effect sizes have been used. Also, if multiple effect sizes have been used, please provide a range for the resulting power values.

We have included the range of power values and effect sizes returned by the SIMR package for Experiment 1 (line 198-199, page 9) and Experiment 2 (lines 549-500, page 24). 

##High vs. low frequency comparison (p11; L 245). Would you mind not estimating statistics regarding differences in stimulus characteristics? This procedure has been identified as problematic, and the problem is described in detail here: Sassenhagen, J., & Alday, P. M. (2016). A common misapplication of statistical inference: Nuisance control with null-hypothesis significance tests. Brain and language, 162, 42-45.

Instead, a value range including how much overlap was present would be more informative.

We have removed the t-test statistics from this section and replaced it with the range of log HAL frequency values for both high and low frequency word sets (lines 233-235, page 10). 

##L12 p. 1: Line sweep missing.

We are not entirely sure what this refers to, but as the first page is the title page we believe this will be resolves when the manuscript is publish in PLOS:ONE

---

## [Editor Report · Decision Letter 2]

25 Jan 2022

The Impact of Hyperlinks, Skim Reading and Perceived Importance when Reading on the Web

PONE-D-21-22263R2

Dear Dr. Fitzsimmons,

We’re pleased to inform you that your manuscript has been judged scientifically suitable for publication and will be formally accepted for publication once it meets all outstanding technical requirements.

Kind regards,

Veronica Whitford, Ph.D.

Academic Editor

PLOS ONE

---

## [Editor Report · Acceptance letter]

26 Jan 2022

PONE-D-21-22263R2 

The Impact of Hyperlinks, Skim Reading and Perceived Importance when Reading on the Web 

Dear Dr. Fitzsimmons:

I'm pleased to inform you that your manuscript has been deemed suitable for publication in PLOS ONE. Congratulations! Your manuscript is now with our production department. 

Kind regards, 

on behalf of

Dr. Veronica Whitford 

Academic Editor

PLOS ONE